# LocDiff: Identifying Locations on Earth by Diffusing in the Hilbert Space

Zhangyu Wang[1,7,2],   Zeping Liu[2],   Jielu Zhang[7,3],   Zhongliang Zhou[3],   Qian Cao[3],
Nemin Wu[3],   Lan Mu[3],   Yang Song[6],   Yiqun Xie[4],   Ni Lao[5,2†],   Gengchen Mai[2†]

[1]University of Maine, [2]SEAI Lab, University of Texas at Austin,
[3]University of Georgia, [4]University of Maryland, [5]Google LLC, [6]Open AI, [7]Harvard University
[†]Corresponding author.

## Abstract

*Image geolocalization* is a fundamental yet challenging task, aiming at inferring the geolocation on Earth where an image is taken. State-of-the-art methods employ either grid-based classification or gallery-based image-location retrieval, whose spatial generalizability significantly suffers if **the spatial distribution of test images does not align with the choices of grids and galleries.** Recently emerging generative approaches, while getting rid of grids and galleries, use raw geographical coordinates and suffer quality losses due to their **lack of multi-scale information**. To address these limitations, we propose a multi-scale latent diffusion model called **LocDiff** for image geolocalization. We developed a novel *positional encoding-decoding* framework called **Spherical Harmonics Dirac Delta (SHDD)** Representations, which encodes points on a spherical surface (e.g., geolocations on Earth) into a Hilbert space of Spherical Harmonics coefficients and decodes points (geolocations) by mode-seeking on spherical probability distributions. We also propose a novel SirenNet-based architecture (**CS-UNet**) to learn an image-based conditional backward process in the latent SHDD space by minimizing a latent KL-divergence loss. To the best of our knowledge, LocDiff is the first image geolocalization model that performs latent diffusion in a multi-scale location encoding space and generates geolocations under the guidance of images. Experimental results show that LocDiff can outperform all state-of-the-art grid-based, retrieval-based, and diffusion-based baselines across 5 challenging global-scale image geolocalization datasets, and demonstrates significantly stronger generalizability to unseen geolocations.

## 1   Introduction

**Location decoding**, i.e., predicting geolocations from given context, remains a challenging and rarely studied problem which has potential applications in numerous tasks including trajectory synthesis [35], building footprint segmentation [14], and the widely studied image geolocalization tasks [38, 45]. As one of the representative tasks for location decoding, image geolocalization aims at predicting locations on Earth based on a given image, which potentially ranges from many types such as wild life photos, street views, and remote sensing images. However, unlike image classification, solutions to image geolocalization are less mature because their ground-truths are locations represented by *real-valued* coordinates on the *spherical* surface. While regression models are commonly used to predict real-valued labels, they are proven to be tricky to train and perform especially poorly on the global-scale image geolocalization problem due to the highly complex and non-linear mapping between the image space and the geospatial space [46, 17]. As an alternative transductive solution, researchers employ pre-defined geographical classes (e.g. divide Earth into

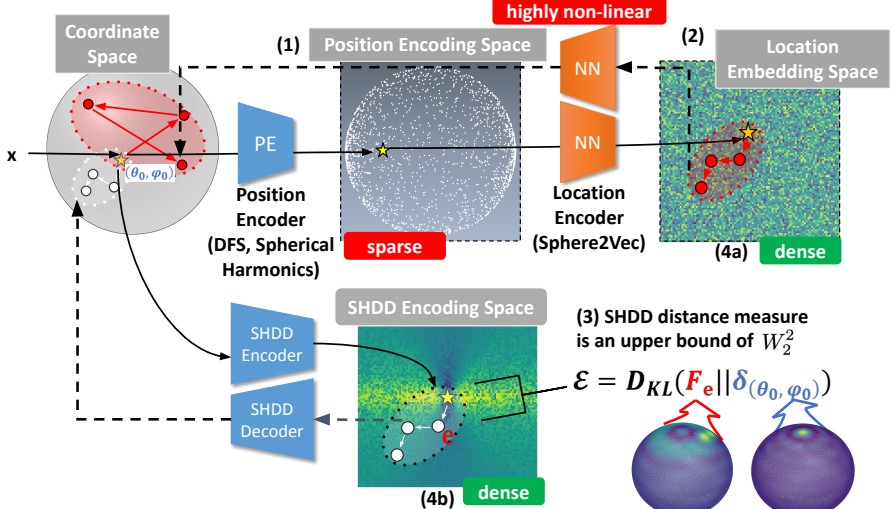

Figure 1: Multi-scale latent diffusion for image geolocalization. **Black solid/dotted arrows** denote encoding/decoding steps. **Orange modules** are learnable, while **blue modules** are deterministic with no learning parameters. **(1)** It is difficult to diffuse in the **position encoding space** [27] because valid positional encodings are sparse which leads to difficulties in diffusion model training and decoding. **(2)** The **locational embedding space** is dense and can perform diffusion processes, but the non-linear mapping between the position encoding and location embedding space makes decoding back to a correct coordinates extremely difficult. Minimizing distances in the location embedding space may not minimize geographic distance. **(3)** The SHDD encoding space is dense – every point $\mathbf{e}$ in this encoding space corresponds to a spherical function $F_{\mathbf{e}}$, whose difference from the spherical Dirac delta function $\delta_{(\theta_0, \phi_0)}$ of the ground truth location $(\theta_0, \phi_0)$ is measured by the reverse KL-divergence $\mathcal{E}$. **(4)** The SHDD decoding addresses the non-linearity problem. The heatmaps **(4a), (4b)** represents the distance from the spherical point represented by the embedding/encoding to the yellow star point in the middle. The distance measured by SHDD is significantly smoother.

disjoint or hierarchical grid cells) [46, 38] or geo-tagged image galleries (e.g. a set of reference geotagged images) [45] to map the real-valued ground-truth coordinates to discrete labels (e.g. the ID of the grid cell or the ID of the reference image in the gallery), subsequently transforming the image geolocalization problem into *a special case of image classification or image-image/image-location retrieval* task. For example, [L]kNN [46], CPlaNet [38], and PIGEON [10] partition the Earth's surface into non-overlapping cells and convert the image geolocalization problem into an image classification problem. GeoCLIP [45] uses a contrastive learning framework to align pretrained image embeddings with geographical location embeddings in the gallery. However, **the spatial resolution of these approaches is constrained by the size of the cells or the availability of gallery images/locations.**

Generative models such as diffusion models [12, 41], score-based generative models [42], and flow matching models [20], have demonstrated great capacity in directly generating continuous outputs such as images and modeling their complex distributions. They are commonly applied to points in Euclidean spaces [42, 12, 41] or the geometric structures defined in Euclidean spaces [49]. This motivates us to develop *diffusion-based image geolocalization methods that decode locations on the spherical surface with a finer spatial resolution by using multi-scale location representations[24, 26, 37] without dependence on predefined grid cells or galleries* [46, 38, 45, 10]. However, simply performing diffusion or flow matching in the original coordinate space faces two major challenges. First, **geographical locations reside on an embedded Riemannian manifold instead of an Euclidean space**[1]. This makes noise adding/removal problematic. If we use coordinates on the manifold such as longitudes and latitudes, the noise adding/removal will lead to projection distortions, especially near the polar areas. If we use coordinates in the 3-D Euclidean space, every forward/backward step will likely output points that are not on the spherical surface [8], which requires noise adding/removal in the tangent space and re-projection back to the spherical surface, such as in Diff $\mathbb{R}^3$ [8] and FM $\mathbb{R}^3$ [8]. The re-projection operation introduces extra computational costs. Secondly, and more importantly, **raw coordinates lack multi-scale information which is necessary for modeling complex spatial distributions** [27, 37]. This practice can yield suboptimal

---

[1]Geographical locations are distributed on a 2-dimensional Riemannian manifold (i.e., the sphere surface) embedded in the 3-D Euclidean space.

geolocalization results. As later shown in Table 2, compared with generative models operating directly on raw coordinates (e.g., Diff $\mathbb{R}^3$, FM $\mathbb{R}^3$ and RFM $\mathcal{S}_2$ [8]), our multi-scale latent location diffusion model LocDiff demonstrates superior performance on 2 image geolocalization datasets across all evaluation metrics measured in various spatial scales.

In order to achieve multi-scale modeling power for complex distributions over space, neural location encoding methods [25, 23, 27, 26, 18, 37, 48, 50] commonly adopt a multi-scale position encoding with deterministic transformations followed by a learnable location embedding layer. However, despite their wide applicability, neither the position encoding nor the location embedding space is suitable for developing a location diffusion model due to **sparsity problem during diffusion** and **non-linearity problem during decoding** as illustrated in Figure 1(1) and (2). Therefore, we hypothesize that

> *The ideal space to develop latent diffusion models for spherical location generation*
> *should be both dense and easy to find projections back to the coordinate space.*

Motivated by this, we propose a novel spherical position encoding method called *Spherical Harmonics Dirac Delta (SHDD) Representation*. Figure 1(3) and (4) illustrates how our method addresses the sparsity problem by encoding a spherical point $(\theta_0, \phi_0)$ as a spherical Dirac delta function $\delta_{(\theta_0,\phi_0)}$. In the SHDD encoding space, every point $\mathbf{e}$ uniquely corresponds to a spherical function $F_{\mathbf{e}}$ and can be seen as an approximation of a spherical Dirac delta function. The level of noise $\mathcal{E}$ can be continuously measured by the reverse KL-divergence between $F_{\mathbf{e}}$ and $\delta_{(\theta_0,\phi_0)}$. Then the latent diffusion in the SHDD encoding space equals gradually adding noise to the ground-truth $\delta_{(\theta_0,\phi_0)}$ (forward process) and finding a sequence of $F_{\mathbf{e}}$ that gradually reduces $\mathcal{E}$ (backward process). During decoding, the learning-free SHDD Decoder evaluates the corresponding spherical function $F_{\mathbf{e}}$ and decodes it as the spherical point whose corresponding spherical Dirac delta function minimizes $\mathcal{E}$. Figure 1 4(b) demonstrates that our SHDD encoding space shows less decoding non-linearity than existing location representation learning methods such as Sphere2Vec [27] and SH [37]. Therefore, diffusion in the SHDD encoding space will be more stable and easier to converge.

Equipped with the Hilbert (i.e. infinite-dimensional Euclidean) SHDD encoding space and the SHDD decoder, we can now **perform conventional latent diffusion for location generation.** We propose a novel SirenNet-based architecture called *Conditional Siren-UNet (CS-UNet)* to learn the conditional backward diffusion process, i.e, to generate spherical points from random Gaussian noise given conditions such as images and texts. We call the integrated framework, including SHDD encoding, CS-UNet latent diffusion, and SHDD decoding, the *LocDiff* model. Results show that LocDiff outperforms all state-of-the-art models on 5 global-scale image geolocalization datasets at large scales (250 km, 750 km, 2500 km), and by combining retrieval-based models (See Appendix A.6), the hybrid variant LocDiff-H achieves superior performance at small scales (1 km, 25 km). LocDiff also outperforms the recently proposed generative geolocalization model RFM $\mathcal{S}_2$ [8], which shows the advantages of our latent diffusion approach over generation in the original coordinate space. Moreover, we demonstrate that LocDiff is more spatially generalizable to unseen locations than retrieval-based models such as GeoCLIP [45].

## 2   Related Work

**Geolocalization by classification and retrieval.**   Traditional geolocalization methods typically employ either a classification approach or an image retrieval approach. The former divides the Earth's surface into non-overlapping or hierarchical cells and classifies images accordingly [31, 46, 30, 10] while the latter approach identifies the location of a given image by matching it with a database of image-location pairs [39, 52, 51, 45]. Using fewer cells results in lower location prediction accuracy while using smaller cells reduces the number of training examples per class and risks overfitting [38]. On the other hand, retrieval-based systems usually suffer from poor search quality and inadequate coverage of the global geographic landscape.

**Geolocalization by diffusion and flow matching in the original coordinate space.**   Dufour et al [8] recently proposed a set of generative image geolocalization models on the original coordinate space, including Diff $\mathbb{R}^3$, FM $\mathbb{R}^3$, and RFM $\mathcal{S}_2$. The first two models perform diffusion and flow matching on the 3D Euclidean space respectively and project the output back on the sphere. RFM $\mathcal{S}_2$, instead, performs flow matching on the spherical Riemannian manifold by projecting back and forth

to the tangent space of each location. However, operating on the original coordinate space cannot capture the rich multi-scale geographic information required for modeling complex distributions on the spherical surface. Our LocDiff overcomes this limitation by performing diffusion in a latent space which captures rich multi-scale location information.

**Riemannian generative models.** Conventional generative models cannot function well in the spherical coordinate space (e.g., 3D coordinates representing points on a sphere) due to the projection distortion and sparsity issue. There are two common strategies to address this problem. The first strategy is to project a point on the sphere to its tangent space (an Euclidean space), add/remove noise in the tangent space, and then spherical re-project the noised/denoised point in the tangent space back to the surface [36] like RFM $\mathcal{S}_2$ [8] does. The second strategy is to derive formulas for direct Riemannian diffusion [15]. The main drawback of both strategies is their computational complexity. In the first strategy, each projection operation takes time, making sampling acceleration based on DDIM [41] impossible, because the projections are accurate only when the diffusion steps are adequately small. For the second strategy, the Riemannian diffusion formulation is much more complicated than the Euclidean version. The model architectures, training tricks, and other useful techniques developed for conventional diffusion models can not be easily transferred.

**Location Embedding.** The distinction between positional encoding and location embedding lies in semantics: the positional encoding is only a task-agnostic transformation of the coordinates $\mathbf{x}$, but the location embedding carries task-specific information. For example, it can contain information about spatial distributions of species if trained on geo-aware species fine-grained recognition tasks [22, 27, 26, 7, 48, 7, 50, 21]. Some prior work, e.g., NeRF [28], utilized positional encoding to represent location information. This task-agnostic method focuses on capturing the position or order of elements within a sequence. In contrast, many location encoders are specifically designed to capture context-aware location information [24]. Please refer to Appendix A.2 and [27] for more details.

# 3 Preliminaries

**Real Basis of Spherical Harmonics** Let $\mathbf{p} = (\theta, \phi)$ be a location on the spherical surface using angular coordinates where $\theta \in [0, \pi)$ and $\phi \in [0, 2\pi)$. For any function $F(\theta, \phi)$ on the sphere, there exists a **unique** infinite-dimensional real-valued vector of coefficients $\{C_{lm}\}$ (we may call it *coefficient vector*) such that $\forall (\theta, \phi), F(\theta, \phi) = \sum_{l=0}^{\infty} \sum_{m=-l}^{l} C_{lm} Y_{lm}(\theta, \phi)$ where $l$ is called *degree* and $m$ is called *order* and $Y_{lm}(\theta, \phi)$ is the *real basis of spherical harmonics* at degree $l$ and order $m$. The detailed computation of $Y_{lm}$ can be found in Appendix A.3. In this way, any function on the sphere can be uniquely represented by its coefficient vector.

**Spherical Dirac Delta Function** Similar to the Dirac delta function in Euclidian space we can define a spherical Dirac delta function is a probability density function over the spherical surface whose mass all concentrates on one point: $\delta_{(\theta_0, \phi_0)}(\theta, \phi) = \begin{cases} \infty & \theta = \theta_0, \phi = \phi_0 \\ 0 & \text{otherwise} \end{cases}$. It can uniquely represent any point $(\theta_i, \phi_i)$ on the sphere by mapping it to $\delta_{(\theta_i, \phi_i)}$. Representing a point as a function allows us to use spherical harmonics to represent points on the spherical surface.

# 4 LocDiff Framework

In this section, we will introduce the theory and techniques we employ in our LocDiff model that enable spherical location generation via latent diffusion. Our aim is to find a position encoding space that does not suffer from the sparsity problem and the non-linearity problem so we can efficiently perform latent diffusion. We first analyze what properties we need to achieve this and propose the Spherical Harmonics Dirac Delta (SHDD) Encoding-Decoding framework accordingly. Then we prove that SHDD satisfies all the desired properties. Following that, we propose the Conditional Siren-UNet (CS-UNet) architecture to learn the conditional backward process for latent diffusion. We also develop computational techniques based on the properties of SHDD representation so that the training and inference of LocDiff are efficient.

### 4.1 Problem Setup and Intuitions

As we have outlined in the introduction, our goal is to find a position encoding method that encodes the spherical surface into a dense subset of $\mathbb{R}^d$ (ideally the entire $\mathbb{R}^d$) and accurately decodes points back to spherical coordinates. There are several mathematical properties such position encoding and decoding method should have. For rigorous discussions, we give definitions of the aforementioned properties and demonstrate how they guide the finding of our SHDD encoding-decoding framework.

**Definition 4.1** (Coordinate Space). A Coordinate Space $\mathcal{C}$ can be any space with a parametrization, such as Euclidean space with the Descartes coordinate system. In this paper, $\mathcal{C}$ always refers to the unit sphere surface embedded in $\mathbb{R}^3$ with the conventional angular coordinate system $(\theta, \phi)$.

**Definition 4.2** (Position Encoding and Position Decoding). A Position Encoder $\mathbb{PE} : \mathcal{C} \to \mathbb{R}^d$ is an injective function, usually $d \gg 3$. $\mathcal{S}_{\mathbb{PE}} := \mathbb{PE}(\mathcal{C}) \subset \mathbb{R}^d$ is called the Position Encoding Space. A Position Decoder $\mathbb{PD} : \mathbb{R}^d \to \mathcal{C}$ is a surjective function.

**The sparsity problem:** Since we are projecting a set of 2-dimensional points in $\mathcal{C}$ into a high-dimensional Euclidean space $\mathcal{S}_{\mathbb{PE}}$, dense filling is impossible.

However, if we define a **difference measure** $\mathcal{E} : \mathbb{R}^d \times \mathbb{R}^d \to \mathbb{R}$, then $\mathcal{S}_{\mathbb{PE}}$ can be *partitioned* by the following equivalence relation:

$$\mathbf{e} \overset{\mathcal{E}}{\sim} \mathbf{e}' \Leftarrow \arg\min_{\mathbf{s} \in \mathcal{S}_{\mathbb{PE}}} \mathcal{E}(\mathbf{e}, \mathbf{s}) = \arg\min_{\mathbf{s} \in \mathcal{S}_{\mathbb{PE}}} \mathcal{E}(\mathbf{e}', \mathbf{s}) \tag{1}$$

that is, we can assign every point $\mathbf{s} \in \mathcal{S}_{\mathbb{PE}}$ to the nearest positional encoding (consequently, a spherical point) in terms of $\mathcal{E}$. We say the $\mathcal{E}$-equivalence classes densely fill $\mathbb{R}^d$. Further, a **learning-free decoder** exists as

$$\mathbb{PD}_{\mathcal{E}}(\mathbf{e}) := \{\mathbf{p} \in \mathcal{C} | \mathbf{e} \overset{\mathcal{E}}{\sim} \mathbb{PE}(\mathbf{p})\} = \arg\min_{\mathbf{p} \in \mathcal{C}} \mathcal{E}(\mathbf{e}, \mathbb{PE}(\mathbf{p})) \tag{2}$$

If $\mathcal{E}$ is **continuous**, i.e.

$$\forall \mathbf{s} \in \mathcal{S}_{\mathbb{PE}}, (\mathbf{e} \to \mathbf{s}) \Rightarrow (\mathcal{E}(\mathbf{e}, \mathbf{s}) \to 0) \tag{3}$$

then the sparsity problem is resolved. Since now diffusion in $\mathcal{S}_{\mathbb{PE}}$ equals a random walk among spherical points and a small perturbation in $\mathcal{S}_{\mathbb{PE}}$ will not result in an abrupt jump on $\mathcal{C}$.

**The non-linearity problem:** Since the diffusion model has intrinsic randomness, it is possible that the generated $\mathbf{e}$ corresponds to a wrong $\mathbf{s}$. If the mapping between $\mathbf{s}$ and its corresponding spherical point $\mathbf{p} = \mathbb{PD}_{\mathcal{E}}(\mathbf{s})$ is highly non-linear (e.g., in the location embedding space), the decoder $\mathbb{PD}_{\mathcal{E}}$ will then be very unstable (see Figure 1). Thus, we hope that for a large tolerance $\eta > 0$ and a small shift $\Delta > 0$, the following property holds for our decoder $\mathbb{PD}_{\mathcal{E}}$:

$$\forall \mathbf{s} \in \mathcal{S}_{\mathbb{PE}}, \mathcal{E}(\mathbf{e}, \mathbf{s}) < \eta \Rightarrow d_{\mathcal{C}}(\mathbb{PD}_{\mathcal{E}}(\mathbf{e}), \mathbb{PD}_{\mathcal{E}}(\mathbf{s})) < \Delta \tag{4}$$

where $d_{\mathcal{C}}$ is the distance in the spherical coordinate space (e.g., the great circle distance). If this property is satisfied, the non-linearity problem is resolved.

It is not an easy task to find such $\mathcal{E}$, especially considering computational constraints (e.g., it is impossible to exactly evaluate the $\arg\min$ function in Equation 1). Fortunately, we find that by treating spherical points as special spherical functions and representing them using Spherical Harmonics coefficients, we can define $\mathcal{E}$ as spherical KL-divergence which satisfies all the desirable properties mentioned above, thus addressing the sparsity and the non-linearity problems as a whole. Moreover, the choice of Spherical Harmonics coefficients also enables efficient computation.

### 4.2 Spherical Harmonics Dirac Delta (SHDD) Encoding

As discussed in Section 3, we can represent spherical points as spherical Dirac delta functions, and a spherical Dirac delta function can be encoded as **an infinite-dimensional real-valued coefficient vector**, i.e. a point in a Hilbert space: $\mathbb{PE}_{\text{SHDD}}(\theta_0, \phi_0) := \bigcup_{l=0}^{\infty} \bigcup_{m=-l}^{l} [C_{lm}]$, where $\bigcup$ denotes vector concatenation. Thus, the spherical harmonics coefficient vector can be used to uniquely represent a point $(\theta_0, \phi_0)$ on the sphere. In practice, we truncate the coefficient vector up to its leading $(L+1)^2$ dimensions, where $L$ is the maximum degree of associate Legendre polynomials:

$$\mathbb{PE}_{\text{SHDD}}^{L}(\theta_0, \phi_0) := \bigcup_{l=0}^{L} \bigcup_{m=-l}^{l} [C_{lm}] \tag{5}$$

We call this vector the $(L+1)$-**degree Spherical Harmonics Dirac Delta (SHDD) Representation** of $(\theta_0, \phi_0)$ and $\mathbb{PE}_{\text{SHDD}}$ the **SHDD encoder**. Each SHDD representation corresponds to an approximation of the true spherical Dirac delta function $\delta_{(\theta_0, \phi_0)}$, whose probability density concentrates in a region surrounding $(\theta_0, \phi_0)$ rather than a single point, enabling differentiable comparison between Dirac delta functions. The Legendre polynomials have finer granularity as their degree $L$ increases, which makes SHDD representations, like other frequency-based location encoding methods such as Sphere2Vec [27], capable of capturing multi-scale spatial information.

Since for spherical Dirac delta functions, the coefficients of the Legendre polynomials are the values of the Legendre polynomials at $(\theta_0, \phi_0)$ [2], i.e.,

$$C_{lm} \equiv Y_{lm}(\theta_0, \phi_0), \forall l, m, \tag{6}$$

the encoding procedure can be reduced to a simple look-up of $Y_{lm}$ values.

### 4.3 SHDD KL-Divergence

Before describing our decoding algorithm we need to measure the difference between any $(L+1)$-degree SHDD representation of a spherical Dirac delta function $\delta$, and an arbitrary $\mathbb{R}^{(L+1)^2}$ vector corresponds to certain spherical function $F$. We propose to use the reverse KL-divergence between (the normalized) $F$ and $\delta$ as the difference measure $\mathcal{E}$.

Let $p_{(\theta, \phi)}$ and $q_{\mathbf{e}}$ be the normalized probability distributions corresponding to the SHDD representation of $(\theta, \phi)$ and an arbitrary $\mathbb{R}^{(L+1)^2}$ vector $\mathbf{e} = \bigcup_{l=0}^{L} \bigcup_{m=-l}^{l} [e_{lm}]$:

$$p_{(\theta, \phi)}(u, v) := \exp\left(\sum_{l=0}^{L} \sum_{m=-l}^{l} Y_{lm}(\theta, \phi) Y_{lm}(u, v)\right) / Z(\mathbb{PE}_{\text{SHDD}}(\theta, \phi)) \tag{7}$$

$$q_{\mathbf{e}}(u, v) := \exp\left(\sum_{l=0}^{L} \sum_{m=-l}^{l} e_{lm} Y_{lm}(u, v)\right) / Z(\mathbf{e}) \tag{8}$$

Here $Z(\mathbf{e})$ is a normalization constant $Z(\mathbf{e}) = \int_0^\pi \int_0^{2\pi} \exp\left(\sum_{l=0}^{L} \sum_{m=-l}^{l} e_{lm} Y_{lm}(u', v')\right) \mathbf{d}u' \mathbf{d}v'$. Then the SHDD KL-divergence is computed by

$$\mathcal{L}_{\text{SHDD-KL}}(\mathbf{e}, \mathbb{PE}_{\text{SHDD}}((\theta, \phi))) := \int_0^\pi \int_0^{2\pi} q_{\mathbf{e}}(u, v) \log \frac{q_{\mathbf{e}}(u, v)}{p_{(\theta, \phi)}(u, v)} \mathbf{d}u \mathbf{d}v \tag{9}$$

It is easy to verify that the SHDD KL-divergence is a continuous difference measure. As for the property described in Equation 4, notice that by [9], the Wasserstein-2 distance $W_2$ between $p_{(\theta, \phi)}$ and $q_{\mathbf{e}}$ is bounded by the KL-divergence in the following inequality:

$$W_2^2(p_{(\theta, \phi)}, q_{\mathbf{e}}) \leq C \mathcal{L}_{\text{SHDD-KL}}(\mathbf{e}, \mathbb{PE}_{\text{SHDD}}((\theta, \phi))) \tag{10}$$

$C$ is a finite constant. $W_2$, being the Earth Mover's Distance, quantifies the amount of probability mass transport between two distributions. Thus, when $\mathcal{L}_{\text{SHDD-KL}}(\mathbf{e}, \mathbb{PE}_{\text{SHDD}}((\theta, \phi)))$ is small, the difference in probability mass distribution is also small, and consequently the largest-mass-region found by the mode-seeking SHDD decoder will also remain mostly unchanged. Figure 1 visualizes this with concrete examples (pretrained Sphere2Vec location encoder and learned neural decoder v.s. our SHDD encoder and decoder) using heatmaps.

### 4.4 SHDD Decoding

**KL-Divergence SHDD Decoder** Following Equation 2, the KL-Divergence SHDD Decoder is:

$$\mathbb{PD}_{\text{KL}}(\mathbf{e}) := \underset{(\theta, \phi)}{\arg\min} \, \mathcal{L}_{\text{SHDD-KL}}(\mathbf{e}, \mathbb{PE}_{\text{SHDD}}((\theta, \phi))) \tag{11}$$

It is impractical to compute $\mathbb{PD}_{\text{KL}}$ exactly. Luckily, Equation 6 makes a natural simplification possible. Notice that minimizing reverse KL-divergence leads to mode-seeking behavior [29], i.e. the $(\theta, \phi)$ that satisfies Equation 11 should fall within the region with the largest probability mass. Thus, we can decode $\mathbf{e}$ by finding the center of its probability mass concentration.

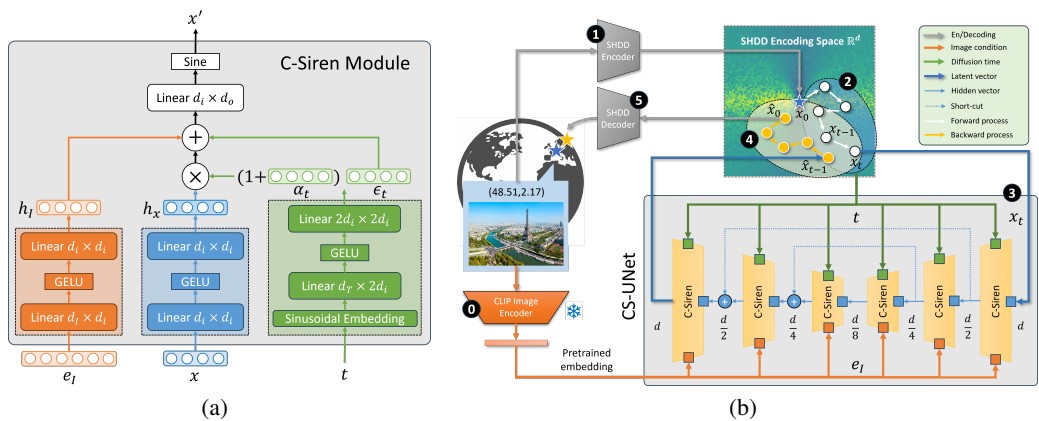

Figure 2: **(a)**: The architecture of Condition SirenNet Module (C-Siren). $x$ is the input latent vector, $x'$ is the output latent vector, $t$ is the scalar timestep, and $e_I$ is the embedding of the input image. $d_i$ is the input dimension, $d_o$ is the output dimension, $d_T$ is the time embedding dimension, $d_I$ is the conditional embedding dimension. **(b)**: The architecture of Conditional SirenNet-Based UNet (CS-UNet) and the workflow of LocDiff. $d$ is the latent dimension. The numbered circles denote the order of training steps.

**Mode-Seeking SHDD Decoder**  Let $\mathbf{e} = \bigcup_{l=0}^{L} \bigcup_{m=-l}^{l} [e_{lm}]$ be an arbitrary vector in $\mathbb{R}^{(L+1)^2}$, then the position decoder $\mathbb{PD}_{\text{mode}}$ is defined as

$$\mathbb{PD}_{\text{mode}}(\mathbf{e}; \rho) := \underset{(\theta, \phi)}{\arg\max} \left\{ \int_{\theta-\rho}^{\theta+\rho} \int_{\phi-\rho}^{\phi+\rho} \exp\left( \sum_{l=0}^{L} \sum_{m=-l}^{l} e_{lm} Y_{lm}(u, v) \right) \mathbf{d}u\mathbf{d}v \right\} \qquad (12)$$

where $\rho$ is a hyperparameter that controls the granularity of the evaluation. There is trade-off between decoding spatial resolution and decoding stability: when $\rho$ is large, we only know the rough range of $(\theta, \phi)$ but the result is less sensitive to local spikes, and vice versa.

One advantage of adopting the SHDD decoder is its **learning-free property**. Unlike learned neural decoders, there is no loss introduced during the decoding stage. Besides, the mapping from diffusion outputs to spherical coordinates is shown to be continuous and relatively smooth. Therefore, it is safe to train latent diffusion models only using the SHDD KL-divergence loss $\mathcal{L}_{\text{SHDD-KL}}$. Another critical advantage is that **the SHDD decoder does not depend on specific partitions of the spherical surface or the spatial distributions of image/location galleries**. This is because the SHDD representation is in effect a continuous spherical function and in theory one can evaluate it in arbitrary resolution. The only two constraints are the maximum degree of Legendre polynomials $L$ which limits the spatial resolution of the spherical function itself and the computational resources (e.g., `float32` or `float64`, evaluation granularity $\rho$), both being independent from other factors.

### 4.5   Conditional SirenNet-Based UNet (CS-UNet)

Inspired by [37], we used SirenNet [40] as the backbone of our diffusion model. The theoretical motivation behind this decision is that Spherical Harmonics coefficients are sums of sinusoidal and cosinusoidal functions (See Appendix A.3). Using sine as the activation function helps preserve gradients because the derivatives of sinusoidal/cosinusoidal functions are still sinusoidal and cosinusoidal functions. Figure 2(a) depicts the network architecture of the Conditional SirenNet (C-Siren) module. The design is straightforward: inputs are the latent vector $x$, the image condition embedding $e_I$, and the diffusion step $t$. First, we use feed-forward layers to project $x$ and $e_I$ into hidden vectors $h_x$, $h_I$. Then we use the sinusoidal embedding layer [41] and feed-forward layers to project the discrete diffusion timestep $t$ into a scale vector $\alpha_t$ and a shift vector $\epsilon_t$. Then, we transform $h_x$ into $h_x = (1 + \alpha_t) \odot h_x + \epsilon_t$, which is an unconditional denoising step. Following that, we sum the transformed $h_x$ and the condition $h_I$ and pass the sum to a feed-forward layer, which adjusts denoising step under the guidance of the condition. Finally, output the sine-activated hidden vector to the next C-Siren module. Figure 2(b) describes the architecture of CS-Unet.

### 4.6 LocDiff

Next, we introduce the training cycle of our LocDiff model as illustrated in Figure 2(b). A training data sample includes an input image $\mathbf{I}$ and its associated geolocation $\mathbf{p} = (\theta, \phi)$ serving as the prediction target. First, we use a frozen CLIP-based image encoder [33] to encode the image $\mathbf{I}$ into an image embedding $e_I$. Then, we encode the geolocation $\mathbf{p} = (\theta, \phi)$ into its SHDD representation $\mathbb{PE}_{\text{SHDD}}(\theta, \phi)$ and store them in a look-up table. Following that, we perform a standard DDPM training [13] based on the proposed CS-UNet architecture as shown in Figure 2(b). In a forward pass of the latent diffusion process, the spherical Dirac delta function $\delta_{(\theta_0, \phi_0)}$ defined by $\mathbb{PE}_{\text{SHDD}}(\theta, \phi)$ will be gradually added noise until being reduced to a vector whose values in each dimension are purely generated from Gaussian noise. In a backward pass of the latent diffusion model, the CS-UNet will start with a noise vector and gradually recover $\delta_{(\theta_0, \phi_0)}$. We implement the DDPM algorithm based on the open-source PyTorch implementation. We use the SHDD KL-divergence $\mathcal{L}_{\text{SHDD-KL}}$ between the ground-truth SHDD representation $\mathbb{PE}_{\text{SHDD}}(\theta, \phi)$ and the diffusion output as the training objective, because it is more computationally stable and preserves the spatial multi-scalability than the spherical MSE (e.g. great circle distance) loss. During inferencing, we sample coefficient vectors from Gaussian noise conditioned on CLIP-based image embeddings and use $\mathbb{PD}_{\text{mode}}$ to predict locations. There are two important implementation details worth mentioning. In practice, the integrals in Equation 9 and Equation 12 are approximated by summation. More specifically, we select a set of $N$ anchor points $\mathcal{A}_N = \{(\theta_i, \phi_i) \in \mathcal{C}\}_{i=1}^N$ on the sphere.

$$\mathcal{L}_{\text{SHDD-KL}}(\mathbf{e}, \mathbb{PE}_{\text{SHDD}}(\theta, \phi)) = \sum_{i=1}^N q_{\mathbf{e}}(\theta_i, \phi_i) \log \frac{q_{\mathbf{e}}(\theta_i, \phi_i)}{p_{(\theta, \phi)}(\theta_i, \phi_i)} \tag{13}$$

$$\mathbb{PD}_{\text{mode}}(\mathbf{e}; \rho) = \arg\max_{(\theta, \phi)} \sum_{i=1}^N \mathbb{I}\{d_{\mathcal{C}}((\theta, \phi), (\theta_i, \phi_i)) < \rho\} \exp\left(\sum_{l=0}^L \sum_{m=-l}^l e_{lm} Y_{lm}(\theta_i, \phi_i)\right) \tag{14}$$

$\mathcal{L}_{\text{SHDD-KL}}$ is used for training, thus we random sample $N = 2048$ anchor points over the globe for each mini-batch to avoid overfitting. As for $\mathbb{PD}_{\text{mode}}$, the choice of $\mathcal{A}_N$ introduces inductive bias – the regions with more anchor points may have heavier impact on the decoding results and higher spatial resolutions. However, Table 3 shows that LocDiff performs stably well on different $\mathcal{A}_N$. See Appendix A.5 for the pseudo codes of LocDiff.

## 5 Experiments

### 5.1 Main Results

We first follow the experimental setup of GeoCLIP [45], a widely used image geolocalization model, for a fair comparison. The training dataset is MP16 (MediaEval Placing Tasks 2016 [19]) containing 4.72 million geotagged images. The test datasets are 3 global-scale image geolocalization datasets – Im2GPS3k [11], YFCC26k [44], and GWS15k [6]. Note that the test datasets Im2GPS3k and YFCC-26k have similar distributions to MP16, and more importantly, their data points might overlap with those in MP16, which benefits retrieval-based approaches [45].

During model inference, we run multiple (16) samplings given different augmentations of each test image and use their geographical center as the prediction. This is a successful strategy proven by GeoCLIP[45] and SimCLR[5]. For example, in our experiments, without this augmentation and averaging step, the 1km accuracy of GeoCLIP on Im2GPS3K may drop from 14% to below 10%. Then we count how many predictions fall into the neighborhoods of the ground-truth locations at different scales respectively, which are denoted as **Street (1 km), City (25 km), Region (200 km), County (750 km), and Continent (2500 km)**. See Appendix A.4, for a detailed ablation study on $L$. Larger $L$ can improve LocDiff resolution, but there is a limit of $L = 47$ before hitting numerical stability issues (Figure 5). We leave numerical stability issue to our future work.

Table 1 summarizes the performance of different models on these three datasets. In addition to the pure LocDiff model, to further increase the resolution without increasing $L$, we develop a hybrid approach denoted as LocDiff-H in Table 1 that combines the advantages of LocDiff and retrieval-based models such as GeoCLIP – we use LocDiff to generate candidate locations, and restrict the retrieval of GeoCLIP to the 200 km radius region around the candidate locations. Please see Appendix A.6 for a detailed description of this hybrid model. From Table 1, we can see that LocDiff-H can outperform

Table 1: Main evaluation results with the GeoCLIP [45] eval setup. $L$ is the degree of SHDD representations used in our model. **LocDiff-H** stands for the hybrid approach of LocDiff plus GeoClip. "Coun." and "Cont." indicate Country and Continent scales. **Bold** and underline numbers denote the best and second best performance on the corresponding dataset. **-** means not reported.

| Model | Im2GPS3k | | | | | YFCC-26k | | | | | GWS15k | | | | |
|---|---|---|---|---|---|---|---|---|---|---|---|---|---|---|---|
| | Street 1 km | City 25 km | Region 200 km | Coun. 750 km | Cont. 2.5k km | Street 1 km | City 25 km | Region 200 km | Coun. 750 km | Cont. 2.5k km | Street 1 km | City 25 km | Region 200 km | Coun. 750 km | Cont. 2.5k km |
| [L]kNN, $\sigma$=4 [46] | 7.2 | 19.4 | 26.9 | 38.9 | 55.9 | - | - | - | - | - | - | - | - | - | - |
| PlaNet [47] | 8.5 | 24.8 | 34.3 | 48.4 | 64.6 | 4.4 | 11.0 | 16.9 | 28.5 | 47.7 | - | - | - | - | - |
| CPlaNet [38] | 10.2 | 26.5 | 34.6 | 48.6 | 64.6 | - | - | - | - | - | - | - | - | - | - |
| ISNs [30] | 10.5 | 28.0 | 36.6 | 49.7 | 66.0 | 5.3 | 12.3 | 19.0 | 31.9 | 50.7 | 0.05 | 0.6 | 4.2 | 15.5 | 38.5 |
| Translocator [32] | 11.8 | 31.1 | 46.7 | 58.9 | 80.1 | 7.2 | 17.8 | 28.0 | 41.3 | 60.6 | 0.5 | 1.1 | 8.0 | 25.5 | 48.3 |
| GeoDecoder [6] | 12.8 | 33.5 | 45.9 | 61.0 | 76.1 | 10.1 | 23.9 | 34.1 | 49.6 | 69 | 0.7 | 1.5 | 8.7 | 26.9 | 50.5 |
| GeoCLIP [45] | 14.1 | 34.5 | 50.7 | 69.7 | 83.8 | 11.6 | 22.2 | 36.7 | 57.5 | 76.0 | 0.6 | 3.1 | 16.9 | 45.7 | 74.1 |
| PIGEON [10] | 11.3 | **36.7** | 53.8 | 72.4 | 85.3 | 10.5 | 25.8 | **42.7** | 63.2 | 79.0 | 0.7 | 9.2 | 31.2 | 65.7 | **85.1** |
| **LocDiff** ($L$=47) | 10.9 | 34.0 | 53.3 | 72.5 | 85.2 | 9.6 | 22.8 | 37.5 | 58.6 | 76.8 | **2.1** | **12.4** | **33.7** | **67.0** | 85.0 |
| **LocDiff-H** ($L$=23) | **15.3** | 36.5 | **56.4** | **75.2** | **87.4** | **13.2** | **26.0** | 41.9 | **64.5** | **80.3** | 0.9 | 7.4 | 33.5 | 66.2 | 85.0 |

all baselines and LocDiff across almost all spatial scales on both Im2GPS3k and YFCC-26k datasets while remaining competitive for the city scale metric on Im2GPS3k and region scale metric on YFCC-26k. On the GWS15k dataset, LocDiff shows the best performance while LocDiff-H underperforms LocDiff, especially on higher spatial scale metrics (e.g., street and city scales). We hypothesize that this is because, unlike Im2GPS3k and YFCC-26k, the spatial distribution of GWS15k is very different from that of the MP16 training dataset. The spatial inductive bias brought by GeoCLIP will not benefit but hurt the performance of LocDiff-H on GWS15k, especially for fine-scale metrics.

Several generative models have been developed recently for image geolocalization including Diff $\mathbb{R}^3$ [8], FM $\mathbb{R}^3$ [8], and RFM $\mathcal{S}_2$ [8]. Since these models were evaluated in different datasets and experimental setups, in order to conduct a fair comparison, we follow the evaluation setup in [8] and compare our LocDiff against these generative geolocalization models on two datasets – OSV-5M [3] and YFCC-4k [46]. The results are shown in Table 2. We can see that LocDiff can outperform all generative geolocalization models on both datasets which clearly showcases the advantages of our

Table 2: Comparison with existing generative methods. The evaluation setup is identical to Dufour et. al [8]. $L$ is the degree of SHDD representations used in our model. **Bold** numbers denote the best performance on the corresponding dataset. **-** means not reported.

| Dataset | Model | City 25 km | Region 200 km | Country 750 km | Continent 2500 km | Density | Coverage |
|---|---|---|---|---|---|---|---|
| OSV-5M | vMF [8] | 0.6 | 17.2 | 52.7 | - | - | - |
| | vMFMix [17] | 0.3 | 11.1 | 34.2 | - | - | - |
| | Diff $\mathbb{R}^3$ [8] | 3.6 | 40.9 | 75.9 | - | 0.752 | 0.568 |
| | FM $\mathbb{R}^3$ [8] | 4.2 | 40.0 | 74.9 | - | **0.799** | 0.575 |
| | RFM $\mathcal{S}_2$ [8] | 5.4 | 44.2 | 76.2 | - | 0.797 | **0.590** |
| | **LocDiff** ($L$=47) | **11.0** | **46.3** | **77.0** | **88.2** | 0.795 | 0.572 |
| YFCC-4k | vMF [8] | 4.8 | 15.0 | 30.9 | 53.4 | - | - |
| | vMFMix [17] | 0.4 | 8.8 | 20.9 | 41.0 | - | - |
| | Diff $\mathbb{R}^3$ [8] | 11.1 | 37.7 | 54.7 | 71.9 | 0.959 | 0.837 |
| | FM $\mathbb{R}^3$ [8] | 22.1 | 35.0 | 53.2 | 73.1 | 1.037 | 0.920 |
| | RFM $\mathcal{S}_2$ [8] | 23.7 | 36.4 | 54.5 | 73.6 | 1.060 | **0.926** |
| | RFM $\mathcal{S}_2$ (10M) [8] | **33.5** | 45.3 | 61.1 | 77.7 | - | - |
| | **LocDiff** ($L$=47) | 33.3 | **46.7** | **65.2** | **79.9** | **1.072** | 0.915 |

multi-scale latent diffusion approach compared with those diffusion models directly applied on the original coordinate space [8].

## 5.2 Generalizability Experiment Results

The key advantage of generative geolocalization over traditional classification/retrieval-based geolocalization methods is that it completely gets rid of predefined spatial classes and location galleries. As noted in [45], the performance of retrieval-based geolocalization methods depends heavily on the quality of the gallery – i.e., how well the candidate locations in the gallery cover the test locations. For example, GeoCLIP uses a 100k gallery with locations drawn from MP16 training data. When using this gallery for the GWS15k dataset, the performance drops due the spatial distribution mismatch between MP16 and GWS15k. As shown in Table 3, GeoCLIP's performance significantly drops when an evenly sampled grid on Earth is used instead of the default image gallery. At small scales, this is explainable because the grids are too coarse to differentiate 1 km to 25 km objects. However, at large scales, the performance of GeoCLIP also drops significantly which is unexpected. With 1 million grid points, the average distance between two candidates is less than 30 km. However, the

Table 3: Generalizability experiment. Numbers outside of the brackets denote geolocalization accuracies. Numbers in the brackets denote relative performance degradation compared to the prior knowledge gallery/anchor. **Bold** numbers denote the largest drop in performance with the given gallery/anchor setting.

| Model | Gallery/Anchor | Size | Street 1 km | City 25 km | Region 200 km | Country 750 km | Continent 2500 km |
|---|---|---|---|---|---|---|---|
| GeoCLIP | MP16 | 100 k | 14.11 | 34.47 | 50.65 | 69.67 | 83.82 |
| | Grid | 1 M | 0.03 ($\downarrow$99.79%) | 9.18 ($\downarrow$73.37%) | 33.47 ($\downarrow$33.90%) | 55.32 ($\downarrow$20.63%) | 75.34 ($\downarrow$10.11%) |
| | | 500 k | 0.03 ($\downarrow$99.79%) | 7.17 ($\downarrow$79.21%) | 29.40 ($\downarrow$41.96%) | 52.29 ($\downarrow$24.94%) | 73.11 ($\downarrow$12.80%) |
| | | 100 k | 0.00 ($\downarrow$100.00%) | 2.67 ($\downarrow$92.25%) | 22.39 ($\downarrow$55.81%) | 47.35 ($\downarrow$32.05%) | 68.77 ($\downarrow$17.94%) |
| | | 21 k | 0.00 ($\downarrow$**100.00%**) | 0.87 ($\downarrow$**97.48%**) | 19.55 ($\downarrow$**61.41%**) | 43.78 ($\downarrow$**37.17%**) | 64.33 ($\downarrow$**23.26%**) |
| **LocDiff** ($L$=23) | MP16 | 100 k | 0.57 | 11.1 | 44.42 | 68.35 | 82.50 |
| | Grid | 1 M | 0.01 ($\downarrow$98.25%) | 4.37 ($\downarrow$60.63%) | 43.04 ($\downarrow$3.10%) | 68.30 ($\downarrow$0.07%) | 81.66 ($\downarrow$1.02%) |
| | | 500 k | 0.07 ($\downarrow$87.72%) | 4.47 ($\downarrow$59.73%) | 43.18 ($\downarrow$2.79%) | 68.36 ($\uparrow$0.01%) | 81.65 ($\downarrow$1.03%) |
| | | 100 k | 0.07 ($\downarrow$87.72%) | 4.04 ($\downarrow$63.60%) | 42.91 ($\downarrow$3.40%) | 68.34 ($\downarrow$0.01%) | 82.18 ($\downarrow$0.39%) |
| | | 21 k | 0.03 ($\downarrow$94.74%) | 4.90 ($\downarrow$55.86%) | 43.44 ($\downarrow$2.21%) | 68.29 ($\downarrow$0.09%) | 81.68 ($\downarrow$0.99%) |

performance of GeoCLIP at the 200 km, 750 km, and 2500 km scales (way larger than 30 km) is still much lower than the performance when using 100K MP16 gallery locations. It indicates that the decline in performance is due to GeoCLIP's weak generalization to new, unseen locations. We can see that the gallery has a strong inductive bias that narrows the spatial scope, and makes the retrieval model easier to overfit, but hurts its spatial generalizability.

Our LocDiff model, while also using anchor points for decoding (training is random), is almost unaffected by the choice of anchor points. To align with GeoCLIP, we compare the same MP16 gallery and evenly sample grid points as decoding anchor points. We can see, at the smaller scales, just like GeoCLIP, introducing the MP16 gallery helps improve the accuracy because its spatial inductive bias helps offset the vagueness of decoding. However, **at larger scales, the performance of LocDiff is almost independent of the choice of anchors** – both the way how we pick the anchor points (MP16 or even grid) and the total number of anchor points (from 21k to 1M). It is a strong indicator of better spatial generalizability for LocDiff.

## 5.3 Computational Efficiency

Though the mathematical derivation of SHDD encoding/decoding seems complicated, our method does **NOT** introduce much computational cost. First, the SHDD encoding/decoding operations are both deterministic, closed-form. For example, the $d$-dimensional ($d = 256, 576, 1024$) SHDD encoding of a point is just a parallel evaluation of a list of two-variable functions, whose time complexity is $O(1)$ and space complexity is $O(d)$. Second, during training, the SHDD encodings can be precomputed and used as an embedding lookup table and there are no further computational overheads. SHDD decoding is implemented as a matrix multiplication followed by an `argmax` operation, which is also very efficient. See Table 4.

| Encoding | | DDPM | | Decoding | |
|---|---|---|---|---|---|
| CPU | 0.0003s per location | GPU | 0.056s per image (200 steps, 16 augmentations) | GPU | 0.0012s per batch (512) |

Table 4: Time efficiency for unit operations in LocDiff

We wish to emphasize that using an SHDD encoded multi-scale representation helps the diffusion process to converge faster, because each dimension encodes information at the respective spatial scales. For example, if at a certain step, the generated SHDD representation is 500km within the ground-truth, then the dimensions with larger than 500km scales will remain mostly intact, and only the finer dimensions need to be altered. In practice, our model takes about 2 million steps to converge on YFCC, while the best results reported in [8] take 10 million steps.

## 6   Conclusion, Limitations and Future Work

In this paper, we propose a novel SHDD encoding-decoding framework that enables multi-scale latent diffusion for spherical location generation. We also propose a CS-UNet architecture to learn conditional diffusion and train a LocDiff model that addresses the image geolocalization task via location generation. It achieves the state-of-the-art geolocalization performance over all existing baselines on 5 global-scale image geolocalization datasets and demonstrates significantly better spatial generalizability. In the future, we aim to leverage the gallery during the diffusion process and explore different ways to further speed up decoding.

## Acknowledgements

This work is mainly funded by the National Science Foundation under Grant No. # 2521631 – A Statistics-Based Geographic Bias Quantification and Debiasing Framework for GeoAI and Foundation Models. Gengchen Mai acknowledges support from the Microsoft Research Accelerate Foundation Models Academic Research (AFMR) Initiative and University of Texas at Austin Library Map & Geospatial Collections Explorer Fellowship. Zeping Liu acknowledges the support of the Amazon AI PhD fellowship. Any opinions, findings, conclusions, or recommendations expressed in this material are those of the authors and do not necessarily reflect the views of the National Science Foundation.

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

# A  Appendix

## A.1  Example of Diffusion Steps

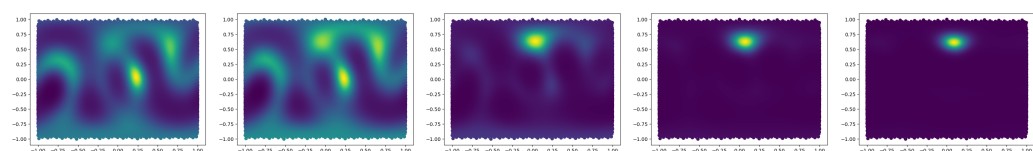

Figure 3: Illustration of how the spherical probability mass concentration (mapped to a plane) corresponding to the SHDD encodings changes along the backward process at step=0, 10, 20, 100, 200, respectively. The more bright, the more probability mass.

## A.2  Sparsity of Existing Positional Encoding Methods

Almost all location encoders can be formulated as the following equation [48]:

$$Enc(\theta, \phi) = \mathbf{NN}(\mathbb{PE}(\theta, \phi)), \tag{15}$$

$\mathbb{PE}()$ is a position encoder that transforms the location $\mathbf{p} = (\theta, \phi)$ into a $W$-dimensional vector, referred to as the position embedding. The neural network $\mathbf{NN}() : \mathbb{R}^W \to \mathbb{R}^d$ is a learnable function that maps the position embedding $\mathbb{PE}(\theta, \phi) \in \mathbb{R}^W$ to the location embedding $Enc(\theta, \phi) \in \mathbb{R}^d$.

**1)** $tile$ is a vanilla location encoder used by many pioneering studies[4, 1, 43]. It divides geographic regions into discrete global grids based on longitude and latitude and learns corresponding partition embeddings based on the grid cell indicator vectors.

**2)** $wrap$ [22] is a sinusoidal location encoder, normalizing latitude and longitude and processing with sinusoidal functions before feeding into $\mathbf{NN}^{wrap}()$, which is composed of four residual blocks implemented through linear layers.

**3)** $wrap + ffn$ [27] is a variant of $wrap$ that substitutes $\mathbf{NN}^{wrap}()$ with $\mathbf{NN}^{ffn}()$, a simple FFN.

**4)** $rbf$ [25] is a kernel-based location encoder. It randomly selects $W$ points from the training dataset as Radial Basis Function (RBF) anchor points. It then applies Gaussian kernels to each anchor points.Each input point $\vec{x}_i$ is represented as a $W$-dimension feature vector using these kernels, which is then processed by $\mathbf{NN}^{ffn}()$.

**5)** $rff$ stands for *Random Fourier Features* [34] and it is another kernel-based location encoder. It first encodes location $\vec{x}$ into a $W$ dimension vector - $\mathbb{PE}^{rff}(\vec{x}) = \varphi(\vec{x})$. Each component of $\varphi(\vec{x})$ first projects $\vec{x}$ into a random direction $\omega_i$ and makes a shift by $b_i$. Then it wraps this line onto a unit circle in $\mathbb{R}^2$ with the cosine function. $\mathbb{PE}^{rff}(\vec{x})$ is further fed into $\mathbf{NN}^{ffn}()$ to get a location embedding.

**6)** *Space2Vec-grid* and *Space2Vec-theory* [25] are two versions of sinusoidal multi-scale location encoders on 2D Euclidean space. Both of them implement the position encoder $\mathbb{PE}(\vec{x})$ as performing a Fourier transformation on a 2D Euclidean space then fed into the $\mathbf{NN}^{ffn}()$. *Space2Vec-grid* treats $x = (\lambda, \varphi)$ as a 2D coordinate while *Space2Vec-theory* be simulated by summing three cosine grating functions oriented 60 degree apart.

**7)** $xyz$ [27] is a vanilla 3D location encoder, converting the lat-lon spherical coordinates into 3D Cartesian coordinates centered at the sphere center with position encoder $\mathbb{PE}^{xyz}(\vec{x})$, then feeds the 3D coordinates into an MLP $\mathbf{NN}^{ffn}()$.

**8)** $NeRF$ can be viewed as a multiscale version of $xyz$ by employing Neural Radiance Fields (NeRF) [28] as its position encoder.

**9)** *Sphere2Vec* [27], including *Sphere2Vec-sphereC*, *Sphere2Vec-sphereC+*, *Sphere2Vec-sphereM*, *Sphere2Vec-sphereM+*, and *Sphere2Vec-dfs*, is a series of multi-scale location encoders for spherical surface based on Double Fourier Sphere (DFS) and *Space2Vec*. The multi-scale representation of *Sphere2Vec* is achieved by one-to-one mapping from each point $x_i = (\lambda_i, \varphi_i) \in \mathbb{S}^2$ with $S$ be the total number of scales. They are the first location encoder series that preserves the spherical surface distance between any two points to our knowledge.

**10)** *Siren (SH)* [37] is a more recently proposed spherical location encoder, which claims a learned Double Fourier Sphere location encoder. It uses spherical harmonic basis functions as the position encoder $\mathbb{PE}^{Siren\,(SH)}(\vec{x})$, followed by a sinusoidal representation network (SirenNets) as the $\mathbf{NN}()$.

These existing location embedding spaces all suffer from sparsity issues, primarily due to the inherent correlations among the different dimensions of the position encoders. The dimensions of position embeddings are frequently interdependent. As a result, many points in the position embedding space become distant or isolated from one another.

### A.3  Computation of Spherical Harmonics

To compute $Y_{lm}$, one can use the following expression in terms of *associated Legendre polynomials* $P_l^m(x)$:

$$Y_{lm}(\theta, \phi) = \begin{cases} (-1)^m \sqrt{2}\,\mathcal{J}\,P_l^{|m|}(\cos\theta)\sin\left(|m|\phi\right) & m < 0 \\ \mathcal{J}\,P_l^{|m|}(\cos\theta) & m = 0 \\ (-1)^m \sqrt{2}\,\mathcal{J}\,P_l^{|m|}(\cos\theta)\cos\left(|m|\phi\right) & m > 0 \end{cases} \tag{16}$$

where $\mathcal{J} = \sqrt{\dfrac{2l+1}{4\pi}\dfrac{(l-|m|)!}{(l+|m|)!}}$ and $P_l^m(x)$ is further computed by

$$P_l^m(x) = (-1)^m \cdot 2^l \cdot (1-x^2)^{m/2} \cdot \sum_{k=m}^{l} \frac{k!}{(k-m)!}x^{k-m}\binom{l}{k}\binom{(l+k-1)/2}{l} \tag{17}$$

### A.4  Spatial Resolution of SHDD Encoding/Decoding

The spatial resolution of SHDD encoding/decoding (i.e., on what scales the mode-seeking decoder can accurately locate the probability mass concentration of the spherical Dirac delta functions) is bound by the degree $L$ of Legendre polynomials. For an $L$-degree SHDD representation, the spatial scale threshold at which it can accurately approximate spherical functions is $\pi/L$ in radian or approximately $20000/L$ in kilometers [16]. For example, for $L = 15, 23, 31$, the thresholds are around 1300 km, 870 km, and 640 km, respectively. At scales significantly below half this threshold, even if the diffusion model generates accurate coefficient vectors, the mode-seeking decoder can still only decode vague locations with large variances. Figure 4 gives a visual intuition.

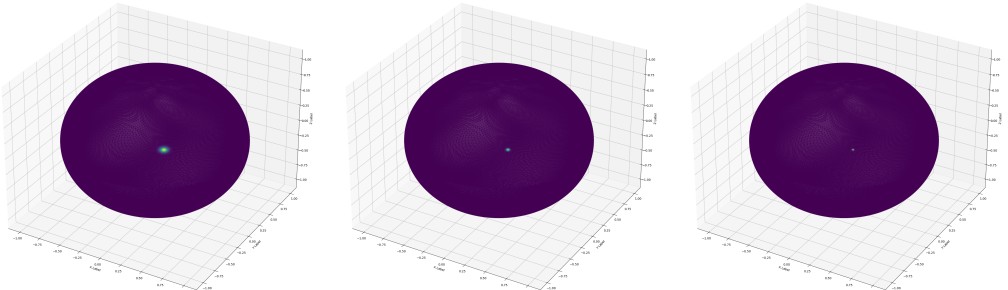

Figure 4: Illustration of the spatial resolutions with $L = 15$, $L = 23$ and $L = 31$. The bright regions are the probability mass concentrations and points within these regions are similarly likely to be decoded as the location predictions. The smaller the bright regions are, the lower errors the SHDD decoding brings.

Therefore, to uplift the performance of LocDiff, one straightforward way is to use larger $L$. We conduct an ablation study of the effect of $L$ on image geolocalization performance on the Im2GPS3K dataset. The results are shown in Figure 5(a). We can see as $L$ increases, while the model performances at larger spatial scales (e.g., 750km, 2500km) only increase slightly, the performances at smaller scales (e.g., 1km, 25km, 200km) see huge uplifting. This validates our hypothesis – *a larger $L$ can make the mode-seeking decoder decoding vague locations with smaller variances, thus leading to higher image geolocalization performance.* The largest $L$ we tried in Figure 5(a) is 47 which corresponds to a spatial resolution of $200km$. This is why we see huge performance improvements on the $25km$ and $200km$ curves but not on the $1km$ curve since $1km$ is still significantly smaller than the current spatial scale threshold.

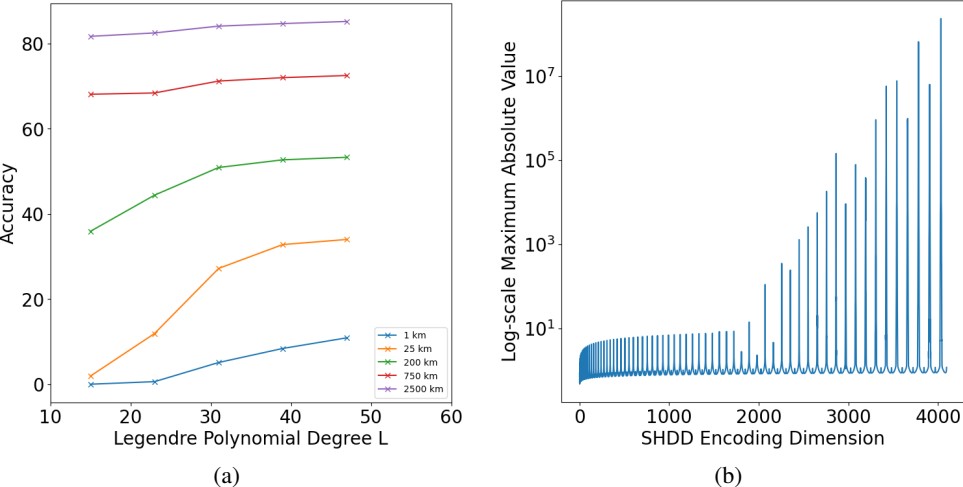

(a)                                        (b)

Figure 5: **(a)**: An illustration of how the image geolocalization performance on the Im2GPS3K dataset increases as L increases. Different curves indicate performance metrics on different spatial scales. **(b)**: A log-scale plot of the maximum absolute values of each SHDD encoding dimension up to 64×64 = 4096 dimensions.

However, it is not recommended to unlimitedly increase $L$. There are two major reasons: The SHDD encoding dimension increases quadratically with $L$, i.e., we need quadratic space to halve the spatial resolution. It is expensive and difficult to train a diffusion model on very large encodings (e.g. to achieve 50 km spatial resolution, we theoretically need 160,000 dimensions). We find that the higher the dimension of SHDD encodings, the higher the maximum absolute values of the coefficients. Figure 5(b) is a log-scale plot of the maximum absolute values of each SHDD encoding dimension up to $L = 63$ (i.e., in total $64 \times 64 = 4096$ dimensions). The absolute values below 2500 dimensions are in general manageable with only a few spikes. However, dimensions beyond this threshold become unbearably large, which makes the probability computation very unstable and easy to overflow. Based on these observations, we use up to $L = 47$ in our paper because now the dimension of SHDD encoding goes to $2304$, still within the manageable range.

Moreover, to address the high dimension issue when we use a large $L$, we find that applying a low-pass filter to the dimensions is a good dimension reduction solution. See Figure 5(b). Many dimensions of the SHDD encodings have very small absolute values and will not significantly influence the results of SHDD encoding/decoding. Thus, we may set a low-pass filter analogous to Fourier transformation and signal processing, which only keeps the dimensions that have adequately large coefficient values. This can be further investigated in future works.

### A.5 Pseudo-Code for LocDiff

---
**Algorithm 1** Training LocDiff
---

**Input** : A dataset $\mathcal{D}$ with location-image pairs $\{(\mathbf{p}, \mathbf{I})\}$. Each location is a tuple of latitude and longitude $\mathbf{p} = (\theta, \phi)$. A Gaussian noise scheduler $\mathcal{N}(t)$, where $t$ is the time step. SHDD encoder $\mathbb{PE}_{\text{SHDD}}$. Pretrained Image encoder $\mathbb{E}_{\text{Im}}$. CS-UNet model $M$ with random initialization. SHDD KL-divergence loss $\mathcal{L}_{\text{SHDD-KL}}$.

**Output** : A trained CS-UNet model $M$.

1  For $\mathbf{p}, \mathbf{I} \in \mathcal{D}$:
    compute the SHDD encoding of $\mathbf{p}$: $\mathbf{e} \leftarrow \mathbb{PE}_{\text{SHDD}}(\theta, \phi)$;
    compute the image embedding of $\mathbf{I}$: $\mathbf{e_I} \leftarrow \mathbb{E}_{\text{Im}}(\mathbf{I})$;
    randomly draw a time step $t$;
    add Gaussian noise to the SHDD encoding (forward process): $\mathbf{e}' \leftarrow \mathbf{e} + \mathcal{N}(t)$;
    use the CS-UNet to denoise the SHDD encoding conditioned on the image embedding (backward process): $\hat{\mathbf{e}} \leftarrow M(\mathbf{e}', \mathbf{e_I}, t)$;
    compute the SHDD KL-divergence loss: $l \leftarrow \mathcal{L}_{\text{SHDD-KL}}(\hat{\mathbf{e}}, \mathbf{e})$;
    use gradient decent to minimize $l$ and update $M$: $M \leftarrow \arg\min_M l$;
2  **return** $M$

---

---

**Algorithm 2** Inferencing LocDiff

---

**Input** : Image $\mathbf{I}$. Random image augmentation $\mathbb{AUG}$. Ensemble number $N$. DDPM sampler $\mathbb{DDPM}$. DDPM step $T$. Pretrained Image encoder $\mathbb{E}_{\text{Im}}$. Trained CS-UNet model $M$ with random initialization. SHDD mode-seeking decoder $\mathbb{PD}_{\text{mode}}$. Mode-seeking range hyperparameter $\rho$. An initial location $\mathbf{p} = (0,0)$.

**Output :** An ensembled location prediction $\hat{\mathbf{p}}$.

3 For $i = 1, 2, \cdots N$:
    generate a random augmentation of image $\mathbf{I}$: $\mathbf{I}_{\text{aug}} \leftarrow \mathbb{AUG}(\mathbf{I})$;
    compute the image embedding of $\mathbf{I}$: $\mathbf{e}_{\text{aug}} \leftarrow \mathbb{E}_{\text{Im}}(\mathbf{I}_{\text{aug}})$;
    randomly draw an SHDD encoding $\mathbf{e}'$;
    use the DDPM sampler and the trained CS-UNet $M$ to sample an SHDD encoding conditioned on the image embedding: $\hat{\mathbf{e}} \leftarrow \mathbb{DDPM}(M, \mathbf{e}', \mathbf{e}_{\text{aug}}, T)$;
    use the SHDD decoder $\mathbb{PD}_{\text{mode}}$ to decode a location from the sampled SHDD encoding: $\hat{\mathbf{p}} \leftarrow \mathbb{PD}_{\text{mode}}(\hat{\mathbf{e}}, \rho)$;
    $\mathbf{p} \leftarrow \mathbf{p} + \hat{\mathbf{p}}$

4 **return** the ensemble of $N$ location predictions: $\mathbf{p}/N$

---

### A.6 A LocDiff-H Hybrid Appoach

We develop a hybrid approach which uses LocDiff's predictions to narrow down the candidate regions and then deploy GeoCLIP to generate the final predictions. More specifically, we first use LocDiff to sample multiple times (e.g., 16) and get a rough distribution of candidate locations, i.e. they indicate where the true answer is highly likely to reside. Then, we restrict the retrieval of GeoCLIP to the neighborhoods (200 km radius) of these candidate locations. As shown in Table 1, we can see that this hybrid approach yields substantially better results than both GeoCLIP and LocDiff alone. This approach is similar to the recommendation systems' retrieve and rerank approach. This flexible hybrid approach points to an interesting future research direction. We can also replace GeoCLIP with other state-of-the-art image geolocalization models such as PIGEON [10], to further improve the model performance.

### A.7 Inductive Bias of Gallery

The key factor that constrains the spatial generalizability of retrieval-based geolocalization models is the inductive bias introduced by the image gallery. When the spatial distribution of the gallery's image locations aligns well with the image locations in the test dataset, the performance of the retrieval-based models will be boosted, especially on low-error scales. However, without such inductive bias (e.g., using evenly spaced grid points as gallery locations), the performance of the retrieval-based models on all scales will suffer.

To better understand what the inductive bias of an image gallery is and how heavily it affects retrieval-based models, we calculate the statistics that demonstrate how spatially aligned the MP16 gallery used in GeoCLIP is with the Im2GPS3K test data. We measure how close test image locations are to the gallery image locations by counting the number of gallery locations that are within 1km/25km from a given test image location. Table 5 shows the statistics results. We can see that the MP16 image gallery's locations indeed closely match the image locations in the Im2GPS3K test dataset. In contrast, when we use a set of grid locations, there are much less locations falling into the 1km or 25 km buffer of the testing image locations.

Table 5: The percentage of test locations that are close (within 1 km/25 km) to multiple gallery locations.

| Gallery | MP16 | | | | Grid | | | |
|---|---|---|---|---|---|---|---|---|
| # Gallery Locations | > 1 | > 10 | > 50 | > 100 | > 1 | > 10 | > 50 | > 100 |
| Within 1 km | 63.5% | 32.7% | 14.9% | 9.78% | 0.1% | 0.0% | 0.0% | 0.0% |
| Within 25 km | 95.2% | 75.7% | 51.9% | 42.0% | 38.9% | 0.0% | 0.0% | 0.0% |

Figure 6 is a set of visualizations of Table 3. It clearly demonstrates how GeoCLIP suffers greatly from using a grid gallery without prior knowledge (i.e., without using the inductive bias brought by the MP16 image gallery), while our method remains almost unaffected on larger spatial scales (200 km, 750 km, and 2500 km) and much less affected on smaller scales (25 km). These results clearly demonstrate that the high performance of GeoCLIP on smaller spatial scales is based on the fact that the MP16 image gallery used by GeoCLIP already contains candidate locations that are close enough to true answers (i.e., test image locations). However, this is not the case for our method because

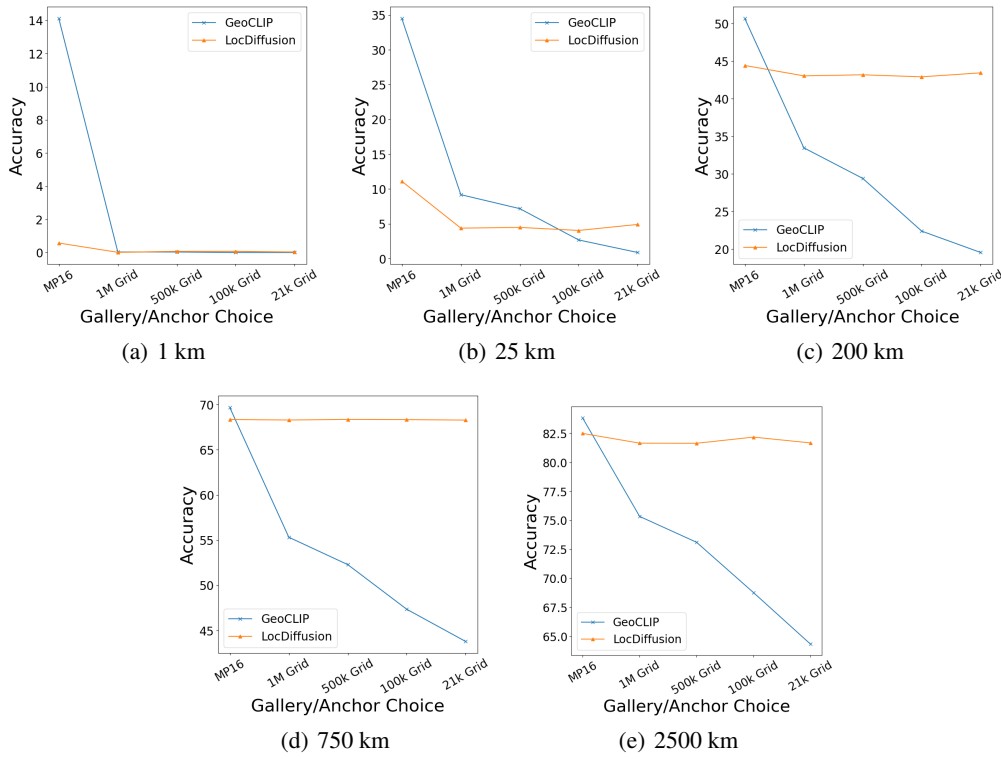

Figure 6: From **(a)** to **(e)**: the performance changes over different choices of galleries/anchor points. Different plots indicate performance metrics under different spatial scales. For each plot, the X-axis indicates the choice of gallery locations, starting from MP16 points, 1 million grid points, to 21K grid points. The y-axis indicates the geolocalization performances on the corresponding spatial scale.

our model does not rely on such an image gallery either during training or during inferencing time. Thus, our LocDiff model suffers much less when we switch to a grid location gallery. Moreover, when we decrease the number of points in the grid gallery, the performances of GeoCLIP decrease significantly while the performances of our LocDiff are almost unaffected.

## A.8 Computational Complexity

We trained our model on a Linux server equipped with four NVIDIA RTX 5500 GPUs, each with 24GB of memory. We report the training time and space complexity on a single GPU in Table 6. We do not have the training times for baseline models such as GeoCLIP and PIGEON because we did not train them from scratch and such statistics are not reported in their papers.

It can be seen that 1024 is the maximum SHDD dimension a single GPU can handle due to GPU memory constraints. For LoDiffusion models with SHDD dimensions beyond 1024, we either use the low-pass filtering technique mentioned in Section A.4 to reduce the dimension to 1024, or split the computation across multiple GPUs. Therefore their computational complexity is not separately reported.

Table 6: Training time and space complexity. Each epoch undergoes 1500 iterations.

| Degree $L$ | Hidden Dimension | Second/Epoch | Memory (MB) |
|---|---|---|---|
| 15 | 256 | 130 | 5691 |
| 23 | 576 | 212 | 10599 |
| 31 | 1024 | 388 | 17407 |

The major factor that decides the inference time of LocDiff is the choice of the sampler. In our experiment, we use the original DDPM sampler (i.e., no DDIM acceleration) with 200 sampling steps. The inference time per image for LocDiff is 0.056s and for GeoCLIP 0.024 seconds.

## A.9 Ablation Studies

### A.9.1 Comparison with other location encoding/decoding techniques

As we have discussed, the superiority of using SHDD for location decoding is that its encoding space is smoother than other location encoders that use neural networks such as rbf and Sphere2Vec [27]. To demonstrate this, we evenly sample 1 million locations on Earth, encode them into corresponding location embeddings by using rbf and Sphere2Vec location encoder, and train a neural network decoder to map the location embeddings back to locations. We also use the learned neural decoder in the LocDiff training with weights frozen. The ablation study results are shown in Table 7. We can see that the performances of rbf and Sphere2Vec are much worse than SHDD, especially on smaller scales. This is because: (1) the learned decoder is not 100% accurate, i.e. it may decode an encoding to a wrong location, and (2) if the encoding gets a small perturbation, the decoded location may have a very large drift due to non-linearity.

Table 7: Comparing the performance of different encoders/decoders on Im2GPS3K. The **NN** is a 6-layer FFN trained on 1 million corresponding location encodings evenly spaced on Earth.

| Encoder | Decoder | 1 km | 25 km | 200 km | 750 km | 2500 km |
|---|---|---|---|---|---|---|
| rbf [24] | NN | 0.0 | 0.0 | 18.2 | 44.1 | 60.2 |
| Sphere2Vec [27] | NN | 0.0 | 0.0 | 22.1 | 58.4 | 72.3 |
| SHDD (L=47) | SHDD (L=47) | 10.9 | 34.0 | 53.3 | 72.5 | 85.2 |

To better understand the spatial drift part, Table 8 shows how much spatial drift will bring to the decoded locations when we add a small Gaussian noise (variance = 0.01) to the corresponding location encoding. We can see that compared with SHDD, both pretrained rbf and Sphere2Vec models can have much larger spatial drifts when we add a small Gaussian noise (variance = 0.01) to the corresponding location encoding. The larger the spatial drift, the less robust the encoding/decoding process is to small hidden space perturbations. Since the diffusion model will not generate perfectly noiseless encodings, such spatial drift indicates the intrinsic error of the corresponding location encoding/decoding method.

Table 8: Comparing the spatial drifts when applying small Gaussian noise (variance = 0.01) to the encoding. The **NN** is a 6-layer FFN trained on 1 million corresponding location encodings evenly spaced on Earth.

| Encoder | Decoder | Perturbation Drift |
|---|---|---|
| rbf | NN | 102.4 km |
| Sphere2Vec | NN | 89.1 km |
| SHDD (L=47) | SHDD (L=47) | 5.3 km |

### A.9.2 Comparison with other losses

Table 9: Comparing the performance of using different training losses on Im2GPS3K.

| Loss | 1 km | 25 km | 200 km | 750 km | 2500 km |
|---|---|---|---|---|---|
| L1 | 0.0 | 0.5 | 20.3 | 30.6 | 43.5 |
| L2 | 0.0 | 0.7 | 20.1 | 32.7 | 44.9 |
| Cosine | 7.5 | 32.2 | 53.0 | 71.5 | 84.9 |
| SHDD (L=47) KL-divergence | 10.9 | 34.0 | 53.3 | 72.5 | 85.2 |

Table 9 shows an ablation study on the impact of different loss functions. We can see that the SHDD KL-divergence is significantly better than L1/L2 losses. Cosine distance, being similar to our SHDD KL-divergence in terms of mathematical formulation (SHDD KL-divergence is the sum of exponential element-wise multiplications, while cosine similarity is the sum of raw element-wise multiplications), has comparable performance especially on larger scales. It would be a good approximation to reduce computational costs. We will add more thorough experiments in the camera-ready version.

### A.9.3 Ablation studies on other modules

We investigate how variations in the width of the CS-UNet affect its performance (see Table 10). In general, shrinking the bottleneck width $w$ of the CS-UNet seems to help alleviate model overfitting (we can adopt a lower dropout rate) and slightly boost performance, but make the model more difficult to train.

Table 10: The bottleneck width $w$ is the narrowest part of each C-Siren module. We report the performance when input encoding dimension is 1024 ($L = 31$) for the sake of limited time.

| Setting | 1 km | 25 km | 200 km | 750 km | 2500 km |
|---|---|---|---|---|---|
| $w = 32, d = 6$ | 5.1 | 27.2 | 50.9 | 71.2 | 84.1 |
| $w = 128, d = 6$ | 4.7 | 27.0 | 50.2 | 70.8 | 84.3 |

### A.10 Training Hyper parameter Setup

Table 11: Training Set-up

| Degree $L$ | Dimensions | | | Hyperparameters | | | | | | |
|---|---|---|---|---|---|---|---|---|---|---|
| | $d$ | $d_I$ | $d_T$ | batch size | lr | epochs | beta | weight decay | dropout | anchor size |
| 23, 47 | 576, 2304 | 768 | 200 | 512 | 0.0001 | 500 | [0.9,0.99] | 0.0005 | 0.3 | 2048 |

Table 11 lists the details of our training setup. We use an Adam optimizer.

