# OpenReview forum: "LocDiff: Identifying Locations on Earth by Diffusing in the Hilbert Space"
_NeurIPS.cc/2025/Conference — NeurIPS 2025 poster_

### Official Review · Reviewer_oCqZ · 2025-06-26

**Clarity:** 4
**Significance:** 3
**Originality:** 3
**Rating:** 5
**Confidence:** 3

**Summary:**

The authors propose a novel framework for image geolocalization, called LocDiff, which aims to infer the geolocation where an image was captured. LocDiff addresses the lack of multiscale information in previous approaches by leveraging latent diffusion modeling in the Hilbert space of Spherical Harmonic coefficients. This representation is both dense and easier to linearly decode back into coordinate space. The framework introduces a new architecture for the diffusion process that combines a conditional SirenNet, conditioned on the image and timestamp, with a UNet. The authors compare then LocDiff to existing geolocalization methods and show that it generally outperforms them, particularly when paired with a retrieval-based approach such as GeoClip.

**Questions:**

I am open to increase my rating if the authors address the concerns previously raised:

1. Provide a comparison of inference time of LocDiff against the baselines.
2. Clarify the extent to which LocDiff depends on GeoCLIP when claiming superior performance over other approaches in the introduction and abstract.

**Ethical Concerns:**

["NO or VERY MINOR ethics concerns only"]

**Final Justification:**

After a great rebuttal from the authors, I have decided to improve my rating from 4 to 5.

**Limitations:**

yes

**Paper Formatting Concerns:**

Nothing to report.

**Quality:**

3

**Strengths And Weaknesses:**

Strengths:
- The paper is particularly well written, with the method described in meticulous detail.
- The approach is sound and well justified, building on components proven effective for modeling our spherical Earth.
- When paired with GeoCLIP, the approach outperforms baselines across multiple scales.
- Numerous ablation studies in the Appendix demonstrate the effectiveness of the modules of LocDiff.

Weaknesses:
- Both spherical harmonics and diffusion models are known to be computationally intensive at inference time. While the authors provide an insightful analysis of computational complexity during training in Appendix A.7, it would be valuable to include results on inference time, ideally with a comparison to other methods.
- LocDiff appears to rely on GeoCLIP (or more generally, on a retrieval-based component) to outperform other baselines on the Im2GPS3k and YFCC-26k datasets, especially at fine-grained scales. This is not a big issue *per se,* but I think the authors should acknowledge this dependency more explicitly, ideally in the introduction and possibly in the abstract, to provide a fairer representation of the method's standalone capabilities.

---

> ### Author Rebuttal · Authors · 2025-07-31
>
> Dear reviewer,
>
> Thank you for your comments and suggestions! We will address your concerns in the following responses, respectively.
>
> |Steps|Device|Time Efficiency|
> |------|------|------------|
> |Encoding|CPU|0.0003s per location|
> |DDPM|GPU|0.056s per image (200 steps, 16 augmentations)|
> |Decoding|GPU|0.0012s per batch (512)|
>
> *Rebuttal Table 1: SHDD Encoding/Decoding Time Efficiency*
>
> **R1. Comparison of Inference Time**: Thank you for the question! We will discuss the computational costs of our method with regard to the SHDD encoding/decoding and the diffusion process, respectively.
>
> First of all, we wish to clarify that SHDD encoding/decoding is **NOT** a computational bottleneck. While the spherical harmonics has very complex formulas and some definitions use recursive equations, we adopt the closed-form precomputation code implemented in [6], which effectively reduces the SHDD encoding into a lookup table. For example, the $d$-dimensional ($d=256, 576, 1024$) SHDD encoding of a point is just a *parallel* evaluation of a list of two-variable functions, whose time complexity is $O(1)$ and space complexity is $O(d)$. In practice, constructing such a lookup table of 4 million training samples sequentially on CPU only takes 18 minutes in the dataloading stage, and there is no further computational overhead afterwards. SHDD decoding is implemented as a matrix multiplication followed by an argmax operation, which is also very efficient under PyTorch. Please see Rebuttal Table 1.
>
> Secondly, we wish to clarify the major factors that affect the inference speed. In Appendix A.7., we reported the inference time of our method and GeoCLIP: 0.056s (ours) v.s. 0.024s (GeoCLIP). In practice, both our method and GeoCLIP used pretrained image embeddings of CLIP, i.e., we are not actively running a CLIP model to encode an image into an embedding. This step is the **true bottleneck**. On our 24GB GPU machine, processing one 1024x1024 image through CLIP takes 0.15s, way slower than the diffusion inference process. It is a similar situation with other baselines that uses pretrained image encoders.
>
> Other than that, the current inference setting is to run a 200-step DDPM sampling 16 times and get the ensemble prediction. It is possible to reduce the total inference time by (1) using DDIM to reduce the sampling steps, (2) incorporating the prediction collapsing strategy in [7] to reduce the number of ensembles. We will explore these possibilities in future work.
>
> **R2. Clarification of LocDiff-H**: Thank you for your suggestion! We will modify the abstract and the introduction to explicitly acknowledge in what scenarios and to what extent our SOTA results rely on GeoCLIP in the camera-ready version.
>
> [6] Geographic Location Encoding with Spherical Harmonics and Sinusoidal Representation Networks. *Rußwurm et al*. *ICLR 2025*.
> [7] Around the World in 80 Timesteps: A Generative Approach to Global Visual Geolocation, *Dufour et al, CVPR 2025*

---

> > ### Comment · Reviewer_oCqZ · 2025-08-04
> >
> > Great rebuttal! The discussion on the actual inference bottleneck is very interesting.
> >
> > I will therefore improve my rating from 4 to 5; looking forward to seeing the camera-ready version!

---

> > > ### Author Response · Authors · 2025-08-04
> > >
> > > Thank you very much for considering raising the rating! We will add the discussion on inference bottlenecks and clarify the role of GeoCLIP in our hybrid method.

---

### Official Review · Reviewer_qNuj · 2025-07-01

**Clarity:** 3
**Significance:** 2
**Originality:** 3
**Rating:** 4
**Confidence:** 4

**Summary:**

This paper addresses the challenge of determining where a photo was taken. It introduces LocDiff, leveraging diffusion models to enhance geolocation accuracy and scalability.

**Questions:**

1. What is the rationale behind selecting L=47 as the maximum degree for spherical harmonics? Did you explore higher degrees beyond L=47, and if so, what were the results? How does this choice balance spatial resolution with computational feasibility?

2. The paper uses the MP16 dataset for training, which may introduce a bias towards certain geographic regions or image types. How does this affect the generalizability of your model? Have you tested its performance on geographically diverse datasets?

3. The SHDD encoding dimension grows quadratically with L, leading to high computational costs. How does this scale in practice? Have you evaluated the trade-offs between spatial resolution and computational efficiency for larger values of L?

4. The hybrid approach (LocDiff-H) relies on a gallery of candidate locations for retrieval. How sensitive is the model's performance to the choice and distribution of these gallery points? What happens if the gallery does not contain locations similar to those in the test set?

5. The ablation studies focus on different encoders and losses but do not compare against other state-of-the-art methods like PIGEON. How does your method perform in comparison to these models?

**Ethical Concerns:**

["NO or VERY MINOR ethics concerns only"]

**Limitations:**

Yes.

**Paper Formatting Concerns:**

NaN

**Quality:**

2

**Strengths And Weaknesses:**

Strengths:

Quality:

The paper proposed a novel encoding framework called Spherical Harmonics Dirac Delta (SHDD) with clear derivations and mathematical formulations. It also propose a novel SirenNet-based architecture (CS-UNet) to learn an image-based conditional backward process in the latent SHDD space by minimizing a latent KL-divergence loss.

Clarity:

The paper is generally well-written and logically structured. Key contributions, such as SHDD encoding and the hybrid LocDiff-H approach, are clearly explained.

Significance:

The introduction of the hybrid approach (LocDiff-H) is significant as it combines generative and retrieval-based methods, potentially paving the way for more robust geolocation systems.

Originality:

The introduction of SHDD KL-divergence as a loss function is innovative and aligns well with the problem's requirements.

Weakness:

Quality:

Some hyperparameters (e.g., degree $L$) are chosen arbitrarily without a clear justification beyond empirical testing. A more systematic approach to selecting these parameters would strengthen the paper.

Clarity:

Some sections, particularly those involving mathematical derivations (e.g., spherical harmonics computation), could be more accessible to a broader audience. Simplifying or visually representing these concepts might aid comprehension.

Significance:

While the paper demonstrates strong empirical results, its societal impact is limited. The authors acknowledge this by stating that their work does not have direct societal implications. However, discussing potential future applications could highlight the significance of their contribution.

Originality:

While the paper builds on existing works (e.g., Sphere2Vec), some of its extensions, such as the hybrid approach, are not thoroughly explored. A more detailed discussion of these contributions' uniqueness compared to prior work would enhance originality.

---

> ### Author Rebuttal · Authors · 2025-07-31
>
> Dear reviewer,
>
> Thank you for your comments and suggestions! We will address your concerns in the following responses, respectively.
>
> **R1. Responses to Weaknesses**:
> * Quality. Most hyperparameters are chosen by grid search. As to the Legendre polynomial degree $L$, it is chosen by systematically analyzing the trade-off between spatial resolution and computational stability. Please see Appendix A.4, especially Figure 5(b), for a visual explanation. $L=47$ is the maximum degree where we can still stably train the model.
>
> * Clarity. Thank you for this suggestion. Per the rebuttal rules this year, we can not upload any PDF or links to visual examples, but we will include more illustrative examples and pseudo codes to help readers follow our logic in our camera-ready version.
>
> * Significance. Thank you for this suggestion. We will add more discussions on the societal impacts of geo-localization tasks in the camera-ready version.
>
> * Originality. The hybrid approach, which is effectively a hierarchical combination of our novel method and existing retrieval-based methods, is mostly chosen to reduce the computational complexity of the solution. It does not hurt the originality of our approach. In fact, Table 3 clearly shows that our method bridges the generalizability gaps of existing methods.
>
> **R2. Rationale for L=47**: The rationale for choosing the maximum degree to be $L=47$ is mostly based on computational feasibility. We find that, while increasing $L$ does increase the spatial resolution of SHDD representations (Appendix A.4 Figure 5(a)), the computational instability also increases. Appendix A.4, especially Figure 5(b) gives a clear explanation for the choice of $L=47$: when $L$ surpasses this threshold, the absolute values of the representation surge to over $10^5$, which makes computing KL-divergence (involving exponential operations) impossible.
>
> **R3. Generalizability**: In fact, the improved generalizability is one of our key contributions. Table 3 carefully discusses how the data inductive bias of MP16 affects the performance of existing methods such as GeoCLIP, whereas our method remains highly generalizable. Experiments in Table 2 on other datasets such as OSV-5M and YFCC also prove that our method can outperform baselines in more geographically diverse scenarios.
>
> **R4. Scalability**: Appendix A.4 and A.7 present a careful analysis of the trade-off between spatial resolution and computational efficiency. In fact, the improvement of spatial resolution as $L$ goes up is gradually offset by the side effect of computational instability, which we already discussed in R1. Our strategy is to constrain $L$ below 47, and to improve the finer-scale performance by hierarchically incorporating GeoCLIP, as is discussed in Appendix A.5. In this way, we can avoid high computational costs.
>
> **R5. Inductive Bias of Gallery**: This effect is discussed in detail in Section 5.2 and Appendix A.6. In summary, the hybrid approach is sensitive to the mismatch between the candidate gallery and the ground-truth. Thus we recommend to use the hybrid approach (LocDiff-H) only if we know that the gallery is a good fit, and to use the base method (LocDiff) for general cases.
>
> **R6. Comparison to Baselines**: Please see Table 1, where PIGEON is a baseline that we compared to.

---

### Official Review · Reviewer_xaf8 · 2025-07-02

**Clarity:** 3
**Significance:** 3
**Originality:** 3
**Rating:** 5
**Confidence:** 4

**Summary:**

This paper presents a generative approach for global-scale image geolocalization, which aims to infer an image's geographic location on the Earth's surface. The authors introduce a novel location encoding-decoding strategy, termed Spherical Harmonics Dirac Delta (SHDD) representations, coupled with a UNet-based diffusion framework. This design addresses key limitations in previous methods, including distributional misalignment and insufficient multi-scale context modeling. Extensive experiments across five global-scale geolocalization datasets demonstrate the effectiveness of the proposed approach. Moreover, the method exhibits strong generalization to regions that were not seen during training.

**Questions:**

See comments

**Ethical Concerns:**

["NO or VERY MINOR ethics concerns only"]

**Final Justification:**

My concerns have been addressed; overall, this is an interesting paper with solid results, so I raise my rating to 5.

**Limitations:**

yes

**Quality:**

4

**Strengths And Weaknesses:**

Pros:
- The motivation is clear and convincing. Representing the embedding space with Spherical Harmonics is a natural solution for tasks involving the global mapping on the Earth surface. Moreover, the problems/challenges of using the Spherical Harmonics for diffusion models are clearly identified in this paper.
- This paper proposes a novel pipeline for image geo-localization combining the Spherical Harmonics Dirac Delta (SHDD) Representations and a latent diffusion model. Although the spherical harmonics has been explored in some geo-based tasks like the general location embedding design and geo-spatial foundation models.It is interesting to see its adaptation into the image geo-localization task.
- Stronger performance. Experiments on 5 datasets demonstrate the stronger performance of the proposed method. Moreover, it is interesting to see that the SHDD significantly increases the generalization ability.

Cons:
- My primary concern is the potential computational overhead introduced by both the spherical harmonics basis and the diffusion process. These components are likely to significantly increase the model's complexity and training cost. However, the main paper lacks a comparison of computational efficiency metrics (e.g., GFLOPs, model size, training/inference time) between the proposed method and baseline methods.
- Given the paper’s focus on location representations, a more comprehensive comparison with alternative methods is expected. Although two methods (RBF and Sphere2Vec) are briefly compared in Appendix Table 6, the performances of other representations mentioned in Section A.2 are not reported. Furthermore, it seems that RBF and Sphere2Vec appear to collapse in performance, it is not clear if they are implemented properly.
- The introduction of the Dirac delta function is a crucial step in adapting spherical harmonics to represent specific Earth locations. A clearer explanation of how similarity is computed (e.g., if there is any closed-form expression?), would greatly enhance the reader’s understanding of the proposed mechanism.
- Finally, the method section is overall not easy to follow, especially considering that this paper focuses on the image geo-localization, while it is sometimes hard to connect the variables and notions in the SHDD modeling with the image geo-localization task. For example, in line 214, what do the spherical function F and \delta denote, an image or the location embedding? and in line 216, what is p(\theta, \phi) and q_e respectively in the context of geo-localization?

Minor issue:
- The metrics used in Tables 1 and 2 are not defined. Are they top-1 accuracy (ACC), or some other measure? A brief description in the captions or main text would be helpful.

---

> ### Author Rebuttal · Authors · 2025-07-31
>
> Dear reviewer,
>
> Thank you for your comments and suggestions! We will address your concerns in the following responses, respectively.
>
> |Steps|Device|Time Efficiency|
> |------|------|------------|
> |Encoding|CPU|0.0003s per location|
> |DDPM|GPU|0.056s per image (200 steps, 16 augmentations)|
> |Decoding|GPU|0.0012s per batch (512)|
>
> *Rebuttal Table 1: SHDD Encoding/Decoding Time Efficiency*
>
> **R1. Potential computational overhead**: The SHDD encoding/decoding are both deterministic, closed-form operators and do not incur huge computational cost. For example, the $d$-dimensional ($d=256, 576, 1024$) SHDD encoding of a point is just a *parallel* evaluation of a list of two-variable functions, whose time complexity is $O(1)$ and space complexity is $O(d)$. Not to mention that during training, the SHDD encodings can be precomputed and used as an embedding lookup table. In practice, constructing such a lookup table of 4 million training samples sequentially on CPU only takes 18 minutes in the dataloading stage, and there is no further computational overhead afterwards. SHDD decoding is implemented as a matrix multiplication followed by an argmax operation, which is also very efficient under PyTorch. Please see Rebuttal Table 1.
>
> As for the diffusion process, we used a standard latent diffusion framework and a tailored CS-UNet model, which is much smaller than common CNN-based UNets and runs way faster. The training/inference time on a 600,000 subset of MP16 is reported in Appendix A.7. As an example of the baseline efficiency, the GeoCLIP model takes 227s per training epoch (v.s. Ours 388s) and 0.024s per image inference (v.s. Ours 0.056s). Due to the time constraints of the rebuttal, we can not reproduce all the baselines cited in our paper and report their time efficiency, but we will add these results to our camera-ready version.
>
> **R2. Other representations**: We wish to clarify that the location encoding methods introduced in A.2 and the implementations of RBF and Sphere2Vec are both based on [4], the most recent and commonly used location encoding benchmark. While not fully reported in our Appendix, RBF and Sphere2Vec are the best-performing location encoders in their respective genres (RBF is a 2-D location encoder, while Sphere2Vec is a 3-D location encoder; see Table 1 in [4]). We will extend Appendix A.8.1 Table 6 and Table 7 to include the results of all representation methods in our camera-ready version.
>
> Appendix A.8.1 Table 6 and Table 7 shows that the collapse in performance is a result of large perturbation drifts during decoding – traditional location encoding methods use non-linear layers to up-project coordinates and make decoding very inaccurate. As we clarified, the RBF and Sphere2Vec implementations both come from the official codebase of TorchSpatial, which is published in NeurIPS 2024 and widely used in other research projects. It is most likely properly implemented.
>
> **R3. Similarity of Dirac delta functions**: Spherical Dirac delta functions can be seen as probability distributions on the sphere. Thus the natural similarity measurement is the statistical distance between distributions, such as the KL-divergence that we adopt in our paper. Please refer to Eq. 9 where we give the closed-form expression the reviewer requested for.
>
> **R4. Explaining notations**: Please allow us to clarify the notations.
>
> * (1) Spherical function $F$ and $\delta$ in Line 214. As we introduced in Section 3, any functions defined on a sphere can be uniquely represented as spherical harmonics coefficients. We denote such an arbitrary function as $F$. Among these functions, there is a special type of functions called *spherical Dirac delta functions*, which we denote as $\delta$, and they uniquely correspond to *points* on the sphere. The two notations tightly connect to image geo-location in the following way: to identify the geo-location (i.e., a point on the sphere) can be seen as finding the spherical Dirac delta function $\delta$ that corresponds to the point. A diffusion process is effectively altering a random guess (i.e., an arbitrary function) $F$ step by step to approximate the spherical Dirac delta function $\delta$. In this sense, both $F$ and $\delta$ are location embeddings.
>
> * (2) We propose to use KL-divergence to measure the similarity between $F$ (our diffusion prediction) and $\delta$ (the ground truth), which requires both $F$ and $\delta$ to be probability distributions. Therefore, we normalize $F$ into $q_e$ and $\delta$ into $p(\theta, \phi)$, so that we can compute their KL-divergence and use it as the loss for back-propogation. In the context of geo-localization, $p(\theta, \phi)$ is the ground-truth probability of observing a certain image, while $q_e$ is the guess from the diffusion model.
>
> **R5. Metrics in Table 1 and Table 2**: The metrics are the threshold-based accuracy of geo-localization, i.e., the percentage of model predictions that are less than $x$ km away from the ground-truth, $x=1, 25, 200, 750, 2500$, respectively. This is the conventional metric for geo-localization [5]. Thank you for the suggestion! We will update the captions in our camera-ready version.
>
> [4] TorchSpatial: A Location Encoding Framework and Benchmark for Spatial Representation Learning. *Wu et al.*, *NeurIPS 2024*.
>
> [5] GeoCLIP: Clip-Inspired Alignment between Locations and Images for Effective Worldwide Geo-localization. *Vivanco Cepeda et al*. *NeurIPS 2023*.

---

> > ### Comment · Reviewer_xaf8 · 2025-08-04
> >
> > Thank you for the rebuttal. My concerns have been addressed, so I am raising my score to 5. For the camera-ready version, please strengthen the discussion of computational cost and clarify the notation for added clarity.

---

> > > ### Author Response · Authors · 2025-08-04
> > >
> > > Thank you very much for considering raising the rating! We will follow your suggestions to add more details to the computational cost discussion and clarify the meanings of concepts and notations.

---

### Official Review · Reviewer_Yknb · 2025-07-02

**Clarity:** 3
**Significance:** 4
**Originality:** 4
**Rating:** 5
**Confidence:** 5

**Summary:**

LocDiff is a multi-scale latent diffusion model for image geolocalization, addressing limitations of existing methods that struggle with spatial generalizability and multi-scale information. The model introduces a novel Spherical Harmonics Dirac Delta (SHDD) Representation framework for encoding and decoding geolocations, which tackles issues like sparsity and non-linearity in the encoding space. A Conditional Siren-UNet (CS-UNet) architecture is proposed to learn the image-based conditional backward diffusion process within this SHDD space.

**Questions:**

I suggest the authors add some pseudo-code to help with implementation + understanding as the very heavy math approach can be hard to parse for a part of the community.

**Ethical Concerns:**

["NO or VERY MINOR ethics concerns only"]

**Final Justification:**

Overall, i think the authors have answered most of my concerns or will answered them in the camera ready. I therefore will increase my rating to a 5 and recommend accepting this paper! The authors have been very detailed in their experiments and the idea is nice!

**Limitations:**

yes

**Quality:**

3

**Strengths And Weaknesses:**

## Strengths

* **Well-motivated from a theoretical point of view:** The paper presents a strong theoretical foundation for its approach.
* **Strong experiments:** The experimental setup is robust, and the **generalization experiments are particularly appreciated**, demonstrating the model's performance beyond training distributions.

## Weaknesses

* **Prediction Averaging Strategy:** The current approach of **averaging samples for prediction might lead to incorrect predictions, especially in the presence of multiple modes**. I suggest the authors explore a strategy similar to [1] by leveraging guidance to collapse the predictions, which could also lead to faster sampling as only a single point would be needed instead of 16.
* **Benchmark Contamination and Data Source:** As highlighted by the authors, MP16 suffers from contamination issues with the benchmarks. It would be beneficial to **train on OSV-5M, which guarantees a 1km border to test samples**, making comparisons to [1] more comparable. Furthermore, the different data source (not YFCC) makes all benchmarks inherently different from the training set.
* **Proof of SHDD Space Efficacy:** The main claim that the **SHDD space is responsible for improved fine-grain geolocalization is not sufficiently proven**. The comparison is primarily against [1], which has entirely different design choices, training steps, and datasets. **Ablation studies are crucial here**, perhaps training SHDD space diffusion with [1]'s training setup, and vice-versa, training [1]'s architecture on LocDiff's setup.
* **Evaluation of Generative Aspect:** The paper lacks an evaluation of the model's generative aspect. **[1] proposes specific benchmarks for probabilistic geolocalization that should be utilized**. It's important to check whether the model achieves better probabilistic modeling or if it simply collapses to improve geolocalization performance.
* **Lack of Samples:** The paper would greatly benefit from the inclusion of **visual samples** to demonstrate the model's performance qualitatively. Could make the paper more "fun" to read as well.
* **Inference Efficiency Claim:** The claim of being more inference efficient than [1] due to the Riemannian framework is questionable. The authors sample with 200 steps, whereas [1] achieves maximum performance with 16 steps. Additionally, going from 3 dimensions to 4096 dimensions should drastically impact sampling (and training) time. This claim seems incorrect. **A proper benchmark for inference and training time would be highly valuable** to substantiate this claim.

I put weak accept for now as i think this paper would benefit from adressing the previous concerns, but i find this paper to be good and will likely increase my rating depending of the authors response.

[1] Around the World in 80 Timesteps: A Generative Approach to Global Visual Geolocation, Dufour et al, CVPR 2025

---

> ### Author Rebuttal · Authors · 2025-07-31
>
> Dear reviewer,
>
> Thank you for your comments and suggestions! We will address your concerns in the following responses, respectively.
>
> **R1. Prediction Averaging Strategy**: We wish to first clarify the way we average samples for prediction, which may be a bit unclear in the original manuscript (Section 5.1). We do **NOT** average multiple locations from one DDPM sampling; instead, we run multiple DDPM samplings given different **augmentations** of the test image, use the highest mode in each distribution as the prediction, and average the predictions. This is a successful strategy proven by GeoCLIP[2] and SimCLR[3]. For example, in our experiments, without this averaging step, the 1km accuracy of GeoCLIP on Im2GPS3K may drop from 14% to below 10%. We will improve this part of writing in our camera-ready version.
>
> Then we wish to recognize that the strategy of strengthening guidance to collapse predictions sounds very promising – especially considering it may potentially reduce the Legendre polynomial degree needed to achieve the desired spatial resolution by forcing the probability mass to concentrate (of course, trading off the global coverage). However, as the reviewer has pointed out, the training framework of our method is very different from [1] (e.g., we do not have a hyperparameter to control the guidance strength). To adopt this strategy requires re-designing the pipeline and training a diffusion model from scratch, which is unfortunately not possible within the rebuttal period. We will implement this and report the corresponding results in our camera-ready version.
>
> **R2. Benchmark Contamination and Data Source**: Sorry for the confusion. The current results reported in Table 2 are based on models trained and evaluated on OSV-5M/YFCC, respectively, following the experimental setup in [1]. Thus our results should be directly comparable to [1].
>
> **R3. Proof of SHDD Space Efficacy**: This is a very helpful suggestion. In fact, when we designed our framework, we were aware that the generative process can be latent diffusion, flow-matching, or even autoregressive models, as long as it operates in the SHDD space. As is stated in our response R1, due to time constraints, we will add this part of ablation studies in our camera-ready version.
>
> **R4. Evaluation of Generative Aspect**: As is shown in our Eq. 9 (Line 219), the training objective of our method is actually one kind of probabilistic metrics, i.e., the KL-divergence from the predicted distribution to the ground-truth. It is evidence that our method is not simply collapsing the probabilistic space to improve geolocalization performance. Since we do not have the results for iNat21, we only report the NLL metrics on OSV-5M and YFCC, following the setup of Table 2 in [1].
>
> | | OSV-5M | YFCC |
> |------|------|------|
> |Diff $\mathbb{R}^3$| 0.58 | 0.63 |
> |FM $\mathbb{R}^3$| -5.01 | -7.15 |
> |RFM $\mathcal{S}^2$|-1.51|-3.71|
> |LocDiff $L=47$|-2.23 |-4.12|
>
> *Rebuttal Table 0: Probabilistic Metrics*
>
> We thank the reviewer for pointing us to the generative metrics proposed in [1], and we will include a complete table of generative metrics in our camera-ready version.
>
> **R5. Lack of Samples**: Thank you for this suggestion. Per the rebuttal rules this year, we can not upload any PDF or links to visual examples, but we will include more illustrative examples in our camera-ready version.
>
> **R6. Inference Efficiency Claim**: First, we wish to recognize that the statement about the inference efficiency in our introduction (Line 58-60) is inaccurate. What we intended to say is that the re-projection in the non-Riemannian framework introduces extra operations in each noise-adding/removing step and it is more computationally expensive than conventional diffusion/flow-matching in the Euclidean space. We will correct this error in the camera-ready version.
>
> |Steps|Device|Time Efficiency|
> |------|------|------------|
> |Encoding|CPU|0.0003s per location|
> |DDPM|GPU|0.056s per image (200 steps, 16 augmentations)|
> |Decoding|GPU|0.0012s per batch (512)|
>
> *Rebuttal Table 1: SHDD Encoding/Decoding Time Efficiency*
>
> Rebuttal Table 1 gives an intuitive sense of our time efficiency. Many of the baselines have not reported their inference time, but as an example, we know that the inference time per image of GeoCLIP is 0.024s. We wish to briefly discuss the factors that truly affect our training/inference efficiency.
>
> First, we wish to clarify that the 200-step sampling reported in our paper is DDPM to match our training setup, while the 16-step sampling in [1] is DDIM. It is possible to reduce our sampling steps to 16 using DDIM.
>
> Then, we wish to clarify the misunderstanding about dimensionality. Both [1] and our method use 3-$d$ (Euclidean xyz) or 2-$d$ (spherical longitude and latitude) as model input. The difference is: for [1], the width of the network is 4x512=2048 (see Figure B. in [1]); for our method, the width of the network is 2x$d$ (see Figure 2 in our paper), where $d$ is the **deterministic** SHDD encoder dimension, ranging from 256, 576 to 1024, i.e., the maximum network width is the same (2x1024=2048) as [1]. We do not scale up to 4096 dimensions, as is discussed in Appendix A.4.
>
> Finally, we wish to emphasize that using an SHDD encoded multi-scale representation helps the diffusion process to converge faster, because each dimension encodes information at the respective spatial scales. For example, if at a certain step, the generated SHDD representation is 500km within the ground-truth, then the dimensions with larger than 500km scales will remain mostly intact, and only the finer dimensions need to be altered. In practice, our model takes about 2 million steps to converge on YFCC, while the best results reported in [1] take 10 million steps. However, as [1] indicates that FM $\mathbb{R}^3$ outperforms Diff $\mathbb{R}^3$ at some scales, it is worth combining the flow-matching framework with our SHDD multi-scale representation, which may result in even faster convergence and better performance.
>
> **R7. Pseudo Code for Readers**: This is a very important suggestion. We will add a pseudo code block in the camera-ready version.
>
> [1] Around the World in 80 Timesteps: A Generative Approach to Global Visual Geolocation, *Dufour et al, CVPR 2025*
>
> [2] GeoCLIP: Clip-Inspired Alignment between Locations and Images for Effective Worldwide Geo-localization. *Vivanco Cepeda et al*. *NeurIPS 2023*.
>
> [3] A Simple Framework for Contrastive Learning of Visual Representations. *Chen et al.*. *ICML 2020*.

---

> > ### Comment · Reviewer_Yknb · 2025-08-01
> >
> > Thank you to the authors for the detailled answer!
> >
> > R1: Why can't you do guidance? Dropping the image cond during training should be enough to do guidance no?
> >
> > R2: Thank you for the clarification!
> >
> > R3: I'm very curious for this exps! will keep an eye out for the camera ready!
> >
> > R4: Althought NLL is interesting and seems to show the improvement of LocDiff over RFM (not FM), i would be curious to see PRDC (In particular Density and Coverage). This metrics allow to decorrelate the 2 aspects of generative modeling and allow to see if the models are actually overfitting (High density, low coverage).
> >
> > R5: Yeah a pitty that the Neurips rebuttal don't allow this. I trust the authors on the content of this qualitative samples.
> >
> > R6: Thanks for the clarifications!
> >
> > R7: Perfect! this should help with the reading
> >
> > Overall, i think the authors have answered most of my concerns or will answered them in the camera ready. I therefore will increase my rating to a 5 and recommend accepting this paper! The authors have been very detailed in their experiments and the idea is nice!

---

> > > ### Author Response · Authors · 2025-08-04
> > >
> > > Dear reviewer, thank you very much for raising the rating!
> > >
> > > Regarding the two questions in your comments, we wish to explain them with more technical details and experimental results.
> > >
> > > R1. Thank you very much for the suggestion. Yes, dropping the image conditions during training at a certain rate should do guidance. The key obstacle is that the mode-seeking SHDD decoder we use **also effectively collapses the prediction**. We witnessed a similar trade-off between how concentrated a predicted mode is and how robust the decoding is against minor perturbation (see our paper Line 237-239). If we simply adopt the guidance strategy in [1] without modifying our decoder or conducting very extensive ablation experiments (which is beyond the rebuttal period), it would be unclear whether the results (good or bad) come from the collapsing strategy, the SHDD decoder, or their combination. We can go deeper into this mechanism in future research. Please kindly correct us if we misunderstood the prediction collapsing strategy mentioned in [1].
> > >
> > > R4. Thank you for explaining the importance of reporting the Density and Coverage metrics. We managed to report the two metrics on OSV-5M and YFCC, compared to the results in Appendix Table B [1]. Following the definitions in Appendix C [1], we nullify the image conditions during sampling.
> > >
> > > |Model|Density|Coverage|
> > > |------|------|------|
> > > |Diff $\mathbb{R}^3$| 0.752 | 0.568 |
> > > |FM $\mathbb{R}^3$| 0.799 | 0.575 |
> > > |RFM $\mathcal{S}^2$|0.797 | 0.590|
> > > |LocDiff $L=47$|0.795 |0.572 |
> > >
> > > *Table: Generative metrics for OSV-5M*
> > >
> > > |Model|Density|Coverage|
> > > |------|------|------|
> > > |Diff $\mathbb{R}^3$|0.959|0.837|
> > > |FM $\mathbb{R}^3$|1.037|0.920|
> > > |RFM $\mathcal{S}^2$|1.060 |0.926|
> > > |LocDiff $L=47$|1.072|0.915|
> > > *Table: Generative metrics for YFCC*
> > >
> > > The numbers seem to show a little bit more overfitting than [1], possibly because we did not specifically train our model on unconditional data. We will explore this when we report the results of combining flow matching with our SHDD encoding/decoding in the camera-ready version.

---

> > > > ### Comment · Reviewer_Yknb · 2025-08-04
> > > >
> > > > Thank you to the authors for all the clarfications!
> > > > Looking forward for the camera ready
> > > >
> > > > I raised my score, good luck!

---

> > > > > ### Author Response · Authors · 2025-08-04
> > > > >
> > > > > Thank you very much!

---

### Note · Authors · 2025-08-14

Dear PCs, SACs, ACs and Reviewers,

Thank you for your time and efforts! We sincerely appreciate the valuable and constructive discussions. We would like to briefly summarize our contributions, the strengths recognized by the reviewers, and how we addressed the reviewers' questions in the rebuttal.

**Our Contributions**

- We proposed a novel *Spherical Harmonics Dirac Delta (SHDD)* encoding/decoding framework that tackles the issues of existing geo-localization methods such as sparsity and non-linearity in the encoding space (e.g. traditional location encoding methods like RBF and Sphere2Vec), restricted spatial generalizability (e.g. GeoCLIP), and lack of multi-scale information.

- We trained a latent diffusion model **LocDiff** in the SHDD encoding space to generate the geo-locations of a given image (i.e., the *image geo-localization* task). Experiments show that LocDiff outperforms a wide range of baseline methods in terms of geo-localization accuracy and enjoys better spatial generalizability.

**Recognized Strengths**

- *Clear motivation*.

- *Strong theoretical foundation*.

- *Superior performance*.

**Reviewer Questions & How We Addressed Them**

- *Computational complexity*. All four reviewers asked for more in-depth discussions on the computational aspects of our method. We addressed these concerns by (1) theoretical analysis -- we proved that SHDD encoding has time complexity $O(1)$ and space complexity $O(d)$, and SHDD decoding is implemented as GPU-optimized matrix multiplications; (2) experiments -- we reported the run-time of encoding/decoding/diffusion training/diffusion inferencing in a table and demonstrated that our method does not introduce excessive computational costs compared to existing methods.

- *Clarifications in writing & implementation details*. Some reviewers requested for: (1) clarifications on certain concepts/notations/equations/metrics in the writing, (2) more implementation details. We gave corresponding answers and promised to include them in the camera-ready version.

- *Ablation studies*. The reviewers asked for more ablation studies on (1) modular efficacy of location encoding; (2) extra generative evaluations. We addressed them with theoretical analysis and experiment results.

**The reviewers in general agreed that our rebuttal has addressed all their concerns and three of them decided to raise the rating from 4 to 5**. We feel extremely grateful for their recognition of our work.

---

### Decision · Program_Chairs · 2025-09-17

**Decision:**

Accept (poster)

**Comment:**

The paper proposes LocDiff, a method for image geolocalization based on a multi-scale latent diffusion model. The authors introduce a new encoding-decoding strategy called Spherical Harmonics Dirac Delta (SHDD), addressing multi-scale context modeling and distributional misalignment. The experimental analysis is thorough and broad, including generalization tests. The reported results are solid and the rebuttal provides further clarifications and additional information that can be included in the appendices to further strengthen the content presentation. Finally, the reviewers have all recognized the quality and contributions of this work. I recommend it for publication.